**Review article**

# Fluid inclusions: tiny windows into global paleo-environments
D. V. Bekaert [1,2] ✉, G. Avice [3] & B. Marty[1]

Geochemical traces of past environments are preserved in the geological record. Although secondary processes often erase this information, fluid inclusions in hydrothermal minerals act as time capsules for reconstructing the evolution of Earth's atmosphere and oceans, including the Great Oxidation Event (GOE). Here, we summarize decades of insights from analyses of ancient fluids in hydrothermal minerals worldwide. These geochemical constraints illuminate the formation of the atmosphere, its evolution through volcanism, escape to space, and subduction. Reconstructions of past atmospheric noble gas and nitrogen compositions, along with ocean salinity, reveal major steps in our planet's evolution. They shed unique light on long-standing questions, including Earth's climate under a faint young Sun, the missing Xe paradox, the cause and timing of oxygenation, the emergence of continents, and the flourishing of life. A refined understanding of the physical mechanisms driving xenon isotopic evolution prior to the GOE may further constrain links between early solar activity and early environmental changes.

Geofluids (waters, hydrocarbons, supercritical fluids, volatile gases) play a crucial role in many geological processes, ranging from molecular-scale fluid-rock reactions to global tectonics, and biological activity within Earth's crust[1]. Fluid inclusions represent remnants of ancient geofluids that have been trapped in hydrothermal minerals (e.g., quartz, baryte). These hydrothermal minerals are typically found in so-called epithermal veins (low to intermediate temperature hydrothermal systems) formed by the intrusion of mineralizing fluids through volcanic, sedimentary, or intrusive units within shallow crustal levels. The distribution of fluid inclusions in hydrothermal minerals depends on several factors, including crystal growth, deformation, and post-crystallization processes. In particular, populations of primary fluid inclusions (formed during crystal growth) are typically aligned along growth zones, trapped in clusters or isolated within the mineral lattice, whereas secondary inclusions preferentially form along healed fractures or microcracks after the mineral has crystallized. Analyzing these inclusions thus provides invaluable information about the geological processes and paleoenvironments in which the minerals formed (e.g., temperature, pressure, and composition). Basically, the volatile element concentrations in any fluid at equilibrium with a gas phase are primarily controlled by Henry's law, where the concentration of a dissolved gas depends on the partial pressure of the gas times a solubility coefficient, which is a function of temperature and salinity[2]. For example, recent analytical developments have demonstrated that Henry's law can be applied to dissolved noble gas concentrations in fluid inclusions from speleothems to reconstruct the cave temperature evolution on millennial timescales, with key implications for paleoclimate studies[3,4].

Fluid inclusions trapped within hydrothermal minerals (e.g., quartz or barite), however, display much more complex histories, with multiple sources and potentially multiple generations of fluid inclusions. Fluid inclusions can be primary, if they were introduced during the formation of the host rocks, or secondary, if they were trapped during subsequent geological events[5]. Assessing the history of these geofluids and understanding the paleoenvironmental information they contain thus invariably requires identifying the timing of formation, the nature, and the composition of their multiple components. In the case of contributions from multiple generations of fluid inclusions, identifying the endmember compositions can be achieved through step-crushing or multiple sample analysis by recognizing chemical and isotopic correlations involving key geochemical tracers (Box Fig. 1).

Elemental ratios (e.g., $N_2/Ar$) may be fractionated by various processes during fluid inclusion formation, presenting a significant challenge in reconstructing the elemental ratios of the paleo-atmospheric component. Importantly, however, isotope fractionation resulting from differences in the solubility of various isotopes is minimal compared to the typical precision of mass spectrometers[6]. As a consequence, the isotope composition of dissolved gases is considered representative of the environment in which the paleofluids circulated. This isotope composition is thus recorded during the formation of fluid inclusions and preserved over geological time (Box Fig. 1). In addition, secondary isotopes (e.g., $^{40}Ar$, $^{136}Xe$) have been/are still being produced in situ by radiogenic and fissiogenic reactions or inherited from paleofluids that circulated through the Earth's crust[7]. The magnitude of these excesses, relative to isotopes that are not significantly produced by

[1]Université de Lorraine, CNRS, CRPG, Nancy, France. [2]Marine Chemistry and Geochemistry Department, WHOI, Woods Hole, MA, USA. [3]Université Paris Cité, Institut de Physique du Globe de Paris, CNRS, Paris, France. ✉e-mail: david.bekaert@univ-lorraine.fr

## Box 1 | Geochemical tools to disentangle hydrothermal and atmospheric components

Overprinting of original fluid inclusion geochemical signals by secondary processes[103] during regional metamorphism episodes may produce modified Archaean seawater (i.e., hydrothermal fluid) signatures that can not be directly used for paleoenvironmental reconstructions. Step-crushing is a geochemical technique that consists in applying sequential crushing under vacuum in order to extract different generations of fluids and tear apart original and secondary (hydrothermal) signals. This approach is preferred to any 'step-heating' method for the study of ancient volatiles in fluid inclusions, as heating may generate undesirable chemical reactions (except for noble gases) that would otherwise not occur within the samples. Step crushing or multiple sample analyses allow one to generate data that are variably influenced by the multiple hydrothermal and atmospheric components contributing to a fluid inclusion population (ref. 8 *and references therein*). Because each extraction step or individual sample will contain varying proportions of each component, the isotopic and elemental compositions of the extracted fluids will show mixing trends whose slopes and intercepts will reflect the proportions of the different components.

Using stepwise crushing, it is also possible to separate different generations of fluid inclusions, especially if the sizes of the corresponding

inclusions are distinct. In the example reported below, we assume a primary generation of fluid inclusions containing a mixture of ancient hydrothermal fluid and ancient ocean water compositions. Over geological time, a second generation of larger fluid inclusion is generated via circulation of secondary hydrothermal fluids. As gentle crushing preferentially affects the main points of fragility in the mineral lattice (i.e., major crystal defects or dislocations, grain boundaries, vacancies), the gas released by gentle crushing, for this example, would be dominated by the larger, secondary fluid inclusions. Only prolongated crushing will allow significantly releasing the smaller, primary fluid inclusions. Varying proportions of ancient hydrothermal fluid and ocean endmembers define a linear correlation (mixing relationship) that allows one to identify the compositions of both endmembers (Box Fig. 1). Accessing volatiles from mineral domains that are unconnected to inclusions or other defects (e.g., radiogenic $^{40}Ar$ produced by in-situ decay of $^{40}K$ within the mineral lattice) requires heating to release the volatiles by diffusion, mineral breakdown, or total fusion. The classical approach, therefore, is to first extract volatiles from fluid inclusions via step crushing, followed by step heating and eventual fusion of the crushing residue[24].

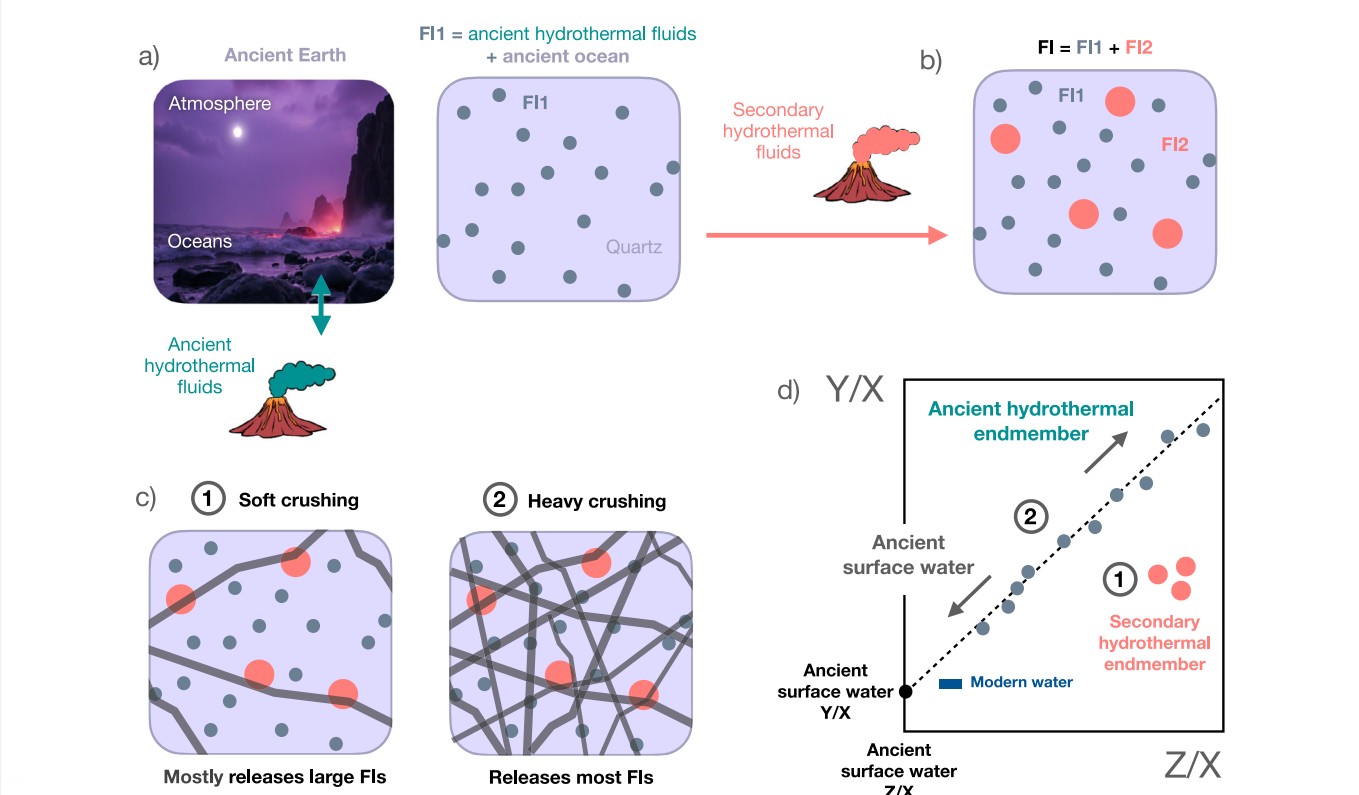

**Box Fig. 1 Illustrated principle of the geochemical deconvolution of hydrothermal and ancient surface component mixing from fluid inclusion step crushing. a** Ancient surface water in contact with the paleo-atmosphere incorporates atmospheric species (e.g., $N_2$, noble gases) through equilibrium dissolution, with potential contribution from hydrothermal fluids. The quantities of dissolved species are determined by gas pressure, fluid temperature, and salinity. **b** Surface waters mix with hydrothermal fluids, which subsequently precipitate as minerals (e.g., quartz, barite), trapping paleofluids as fluid inclusions (FIs). For the sake of illustration, we consider two generations of FIs (one primary and one secondary), with second generation FIs being larger than primary ones. Note that this is however just an example, as there is no clear systematics between the timing of a hydrothermal event and the size of the corresponding FIs. Such a description of fluid inclusions is intentionally oversimplified, as fluid inclusions in hydrothermal quartz grains are typically less than a few tens of micrometers in size. Consequently, a single mineral grain may contain several generations of fluid inclusions with distinctly different compositions that may not be easily separable during the crushing process. **c** In the laboratory, dissolved gases are extracted for analysis of their concentrations and isotopic compositions. Different generations of fluid inclusions may be extracted via step crushing. **d** In most cases, the end-member components of trapped fluids can be identified using specific geochemical tools (e.g., multicomponent mixing deconvolution), enabling the identification of the paleo-surface water endmember. This information provides insights into paleo-environmental parameters, including the composition and pressure of the paleo-atmosphere, as well as the salinity of the paleo-oceans.

secondary processes (e.g., $^{36}Ar$, $^{130}Xe$), can be quantified and potentially used as a tool to date the fluid inclusions (e.g., Ar-Ar geochronology[8]) and place time constraints on reconstructed paleoenvironments.

Over the past decades, many studies have attempted to peer through the veil of volatile compositions in fluid inclusions in hydrothermal minerals worldwide to reconstruct the evolution of Earth's atmosphere and oceans over time, including major events such as the Great Oxidation Event (GOE), around 2.4 to 2.0 billion years ago[9,10]. As of today, the cause(s) and exact timing of the GOE remain debated[11]. The appearance of oxygenic photosynthesis, the reduction of methanogen activity, and drastic changes in tectonic activity (including continent weathering rates) and volcanic activity are all factors that could have contributed to or modulated the rise of $O_2$ in the atmosphere, whether it occurred gradually or suddenly[12]. Over geological time, the composition of the atmosphere results from a balance between outgassing of the solid Earth (mantle and crust) and ingassing into the solid Earth via subduction[13,14]. While volcanoes are a major pathway for outgassing, especially for $CO_2$, other volatiles—including noble gases—are also significantly released to the atmosphere during cryptic degassing, erosion, and metamorphic processes[15]. In a net outgassing regime, the inventories of volatiles at Earth's surface (assuming no escape to space) will increase with time. Varying the inventories of nitrogen and halogens at Earth's surface could, for example, have implications for the total atmospheric pressure and ocean salinity, respectively, with key implications for the climate and the development of life. Because the isotopic and elemental composition of the degassing mantle is distinct from that of Earth surface, the composition of the atmosphere and oceans is also expected to change through time[15]. Unraveling the timing and extent of these changes in the volatile content of the Earth's surface inventory has implications for understanding long-term interactions between our planet's interior and surface, and their implications for the evolution of surface environments. In this contribution, we summarize the wealth of information that has been gained from the analysis of fluid inclusions trapped in ancient minerals on the formation and evolution of our planet's atmosphere and oceans, ultimately leading to the establishment of a hospitable environment for life.

## Noble gases

Noble gases are considered inert under most terrestrial conditions, meaning they are not influenced by chemical or biological processes. Under ionized conditions or at high temperature/high pressure in the deep Earth, however, the enhanced reactivity of heavy noble gases (especially Xe) makes it possible for these elements to be incorporated into molecules and mineral phases[16,17]. Under neutral ambient conditions at Earth' surface, the concentrations and isotope compositions of noble gases dissolved in paleo-fluids are controlled solely by physical and nuclear processes, which can be modeled and quantified[18]. Although the lightest noble gases, helium and sometimes neon[19], have shown a

propency for diffusive loss, fluid inclusions are generally retentive of heavier noble gases, including argon, krypton, and xenon.

## Atmospheric xenon isotopes as a tracer of Earth's surface oxidation

One of the most remarkable observations that has come from the analysis of noble gases contained in fluid inclusions trapped in ancient minerals is that Archean atmospheric Xe presents an isotopic composition intermediate between the one of modern atmospheric xenon and that of xenon from planetary precursors referred to as U-Xe[20–24] (Fig. 1). When combined together, ancient atmospheric Xe data point towards a global and protracted evolution of atmospheric Xe isotopes via mass-dependent fractionation (MDF, i.e., a process by which isotopes of an element are separated or fractionated according to their mass differences), presumably from the Hadean until the late Archean[25]. In contrast, Kr isotope signatures in ancient atmosphere samples have consistently been found to be indistinguishable from modern composition. To have a higher degree of MDF for Xe despite Kr being a lighter element is unexpected, suggesting that a Xe-specific process is required to account for atmospheric Xe MDF throughout the Archean. Because Xe has a low ionization threshold (12.13 eV) relative to the other noble gases (e.g., 15.76 and 14.00 eV for Ar and Kr, respectively; making it easier to remove an electron from its outermost shell), atmospheric Xe could have been readily ionized by enhanced ultraviolet radiation. Ionized Xe could then have then been being dragged along open magnetic field lines and lost to space via ionic coupling with escaping $H^+$[26] or becoming trapped in organic hazes possibly formed within the $CH_4$-rich early atmosphere[27]. One major challenge of this atmospheric Xe escape model is that vertical atmospheric transfer is required to sustain the upward transport of Xe ions through the molecular ionosphere to the base of the outflowing hydrogen corona and to maintain Xe escape[26]. The nature and timescales of the mechanisms driving such vertical transfer remain to be described. While the exact mechanisms behind Xe isotope evolution over time are still being explored, it is understood that this evolution was global, prolonged, and reflects long-term atmospheric changes.

Variations in atmospheric Xe isotopes through the Archean period appear to have stopped around the GOE, similar to sulfur mass-independent fractionation (S-MIF) signals[25]. Although it is not possible to firmly establish whether Xe isotope evolution followed a linear, exponential, or power law decay throughout the Archean[23], these variations may provide insights into past history of continental crust build up[28] and ancient levels of oxygen ($O_2$), methane ($CH_4$), and hydrogen ($H_2$) in the atmosphere[29]. In particular, the Xe isotope trend is thought to relate to how much hydrogen escaped from Earth, serving as a potential indicator of Earth's overall oxidation over time[30]. The progressive mass fractionation of xenon is best explained by xenon being dragged into space by escaping hydrogen during the Hadean and Archean eons[26]. Xenon ions could have been carried away by hydrogen escaping

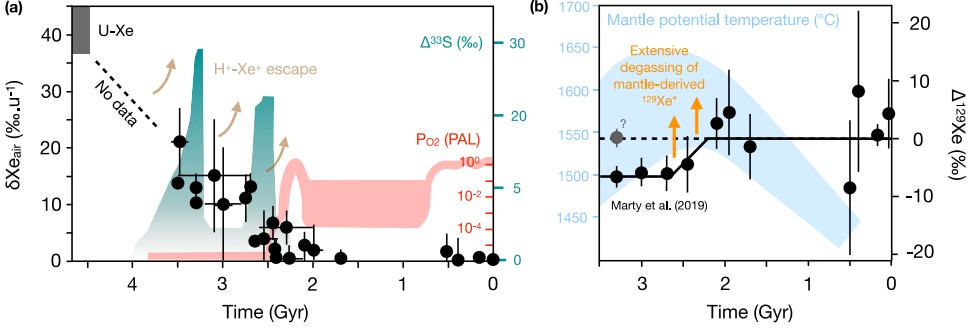

**Fig. 1 | Evolution of the isotopic composition of atmospheric Xe over time. a** The isotopic evolution of atmospheric Xe is shown along with the evolution of mass-independent sulfur isotope signals (MIF), and atmospheric oxygen concentration. This evolution of atmospheric Xe has been primarily derived from the analysis of fluid inclusions in hydrothermal minerals[22]. Whether the evolution of atmospheric Xe was continuous or occurred in steps remains debated[24]. **b** Time evolution of the deficit of $^{129}Xe$ ($\Delta^{129}Xe$) in ancient atmospheric gases relative to the modern atmosphere, compared to petrological estimates of mantle potential temperature (TP) for non-arc lavas[60]. Recent reanalysis of the 3.3 Gyr-old Barberton quartz revealed no apparent $^{129}Xe$ deficit (see gray dot noted with a question mark), indicating that more data is needed to assess the magnitude and timing of $\Delta^{129}Xe$ evolution in the ancient atmosphere[43].

Earth's atmosphere through strong polar winds, which required significant hydrogen from $CH_4$ and/or $H_2$ in the lower atmosphere. As such, the isotopic evolution of atmospheric Xe serves as a direct proxy for hydrogen escape from early Earth, a process that would tend to oxidize Earth's surface[30], until the rise of free atmospheric $O_2$[25,29]. Models also suggest that at least 1% of total hydrogen (e.g., stored in $H_2$ and $CH_4$) must have been present to facilitate xenon escape[26]. Considering independent constraints that set an upper limit of $10^{-2}$ bar for the partial pressure of $H_2$, this suggests that a significant portion of hydrogen was stored in methane[30,31]. Crucially, this Xe escape scenario is only possible in an anoxic atmosphere, as oxygen-rich (oxic) air destroys $H_2$ and $CH_4$, reducing the hydrogen needed to drag xenon out of the atmosphere. Additionally, in an oxygen-rich environment, $O_2$ would remove Xe ions through efficient resonant charge exchange reactions[26].

According to recent modeling by Zahnle et al.[26], the prolonged period of Xe escape suggests that, if this process were constant, Earth would have lost hydrogen equivalent to at least one ocean's worth of water, roughly divided between the Hadean and Archean eons. These authors concluded that Xe escape must have rather occurred through small openings or in brief episodes, which appears consistent with growing evidence that Xe escape was not a continuous process[24]. Although the characteristics of Earth's magnetic field during the Hadean and Archean eons remain uncertain[32], one possibility is that Xe escape was confined to polar regions due to opening of the geomagnetic field, driven by sporadic bursts of intense solar activity, restricted to short periods with abundance hydrogen, or a combination of all these factors. Although the exact mechanism (or combination of mechanisms) controlling atmospheric Xe escape from the ancient atmosphere remains to be understood, the isotopic evolution of atmospheric Xe could represent a new geochemical proxy of past solar activity and geomagnetic field variations, which both remain largely uncertain[33,34]. Interestingly, atmospheric xenon on Mars also presents a mass-dependent enrichment in heavy Xe isotope[35]. The extent of Xe isotope fractionation in the Martian atmosphere (2.5%/amu relative to a solar initial composition) has long been considered smaller than that on Earth (~3.8%/amu relative to U-Xe[36]). However, more recent estimates of present-day terrestrial atmospheric Xe isotope fractionation, relative to its initial precursor, revise this value down to 2.6%/amu, comparable to that of Mars[37]. It was historically proposed that Earth could have formed from Mars-like embryos, and that certain characteristics of the present-day atmosphere (e.g., atmospheric Xe isotope fractionation) could reflect processes that occurred on these early building blocks[38]. However, measurements of trapped Martian atmospheric xenon in Martian meteorites also suggest that the isotopic fractionation of atmospheric xenon was still ongoing 4.4 Ga ago, but that the final amplitude was established as soon as 4.1–4.2 Gyr ago[39]. Recently, Shorttle et al.[40] proposed that energetic collisions that happened on early Mars (200–300 Ma after solar system formation) were powerful enough to drive atmospheric escape of xenon. It remains unknown if this process was also efficient on the Hadean Earth. Zahnle et al.[26] also noted that the relative Xe depletion differs between the two planets, with Xe in the Martian atmosphere being ~50% less depleted than in Earth's atmosphere. This suggests that the Xe escape process operated differently on the two planets, and that gravity is not the dominant factor in atmospheric escape (as Mars, being smaller, would be expected to have lost more Xe). Instead, this points to a non-thermal escape process involving ion loss. As detailed below, the scenario of atmospheric Xe escape is consistent with the fact that Earth's atmosphere has a Xe/Kr ratio that is 10 to 20 times lower than expected from its cosmochemical precursors, suggesting selective Xe loss[21,22,41–43].

## Missing atmospheric Xenon

Over the past decades, several hypotheses have been proposed to account for the apparent missing Xe via sequestration into various terrestrial reservoirs (including shales[44], ice[45], the continental crust[46], and the Earth's mantle[28,47] and/or core[48]; Box 2). However, these hypotheses face great challenges in accounting for the depletion of Xe relative to Kr in Earth's atmosphere and,

most critically, the isotopic evolution of atmospheric Xe throughout the Archean (Fig. 1). That the analysis of noble gases trapped in Archean rocks revealed no significant evolution in atmospheric Kr isotopes over time is consistent with the fact that only Xe eventually went missing[49].

Atmospheric isotopes can be fractionated by escape to space due to differences in mass between isotopes, which affect how they interact with physical processes that drive escape. In the case of thermal escape (also referred to as Jeans Escape[50]), lighter isotopes, having lower mass, can reach the escape velocity of a planet more easily than heavier isotopes. In the upper atmosphere, where gas particles have higher kinetic energy, lighter isotopes are more likely to achieve the velocity needed to escape the planet's gravitational pull. Over time, this selective escape process leads to the fractionation of isotopes, with a higher proportion of lighter isotopes being lost to space. Non-thermal escape mechanisms refer to processes for which charged particles are involved, like photochemical reactions, solar wind interactions, and polar outflow[51], which can also cause isotope fractionation. During periods of intense solar radiation, hydrogen escape can create a drag effect, pulling along heavier isotopes in a process known as hydrodynamic escape. However, lighter isotopes still tend to escape more efficiently due to their lower mass. All these mechanisms cause the remaining atmosphere to become rich in heavier isotopes, while lighter isotopes become relatively depleted, resulting in isotopic fractionation. Thus, explaining the Xe isotope fractionation in the ancient atmosphere, while not observing isotopic fractionation in lighter noble gases through conventional atmospheric escape processes, presents a significant and complex challenge.

Unlike lighter Ar and Kr, Xe is easily ionized by solar UV or charge exchange with $H^+$ ions, so $Xe^+$ can be dragged out to space by escaping $H^+$ ions without significantly affecting atmospheric Kr. Unlike Xe, Kr ions are neutralized by reaction with $H_2$[52], therefore explaining the lack of Kr isotope fractionation in the ancient atmosphere. As such, the scenario of atmospheric Xe escape to space provides a straightforward explanation for the longstanding missing Xe paradox[42]. Recently, noble gas data from 3.0 Gyr-old Barberton (South Africa) quartz fluid inclusions suggested that the Xe/Kr ratio was higher in the Archean than it is today[43], consistent with the selective and prolonged escape of atmospheric Xe. More data is needed to establish how the Xe/Kr ratio in the atmosphere evolved over time until reaching its present-day 10- to 20-fold Xe depletion.

## Tracking mantle degassing using mono-isotopic noble gas excesses

Radiogenic $^{40}Ar$ has been produced within Earth's silicate reservoirs through the radioactive decay of lithophile $^{40}K$ ($T_{1/2} = 1.25$ Ga; Box 3). Earlier studies modeled the evolution of atmospheric argon's isotopic composition (specifically the $^{40}Ar/^{36}Ar$ ratio) due to long-term degassing of radiogenic argon into the atmosphere[53,54]. The discovery in the late 1970s that the 400-million-year-old atmosphere had a slightly lower $^{40}Ar/^{36}Ar$ ratio than the modern atmosphere paved the way for paleo-atmospheric studies related to Earth's mantle and geodynamics[55,56]. Initial data from the Archean atmosphere seemed to confirm that Earth's crust, through potassium storage, played a significant role in modulating the atmospheric $^{40}Ar/^{36}Ar$ ratio[54,57] (Fig. 2). Recently, a model coupling He, Ne, and Ar isotope systematics suggested that mantle outgassing plays a prominent role in controlling the outgassing of radiogenic $^{40}Ar$ into the atmosphere, while the role of the continental crust remains minor[58].

Similar to $^{40}Ar$, the mantle is enriched in $^{129}Xe$ compared to the atmosphere, due to the radioactive decay of $^{129}I$ during the first ~100 million years of Earth's evolution[59]. Over time, volcanic gas was continuously outgassed from Earth's interior, leading to a progressive buildup of $^{129}Xe$ in Earth's atmosphere relative to other Xe isotopes (Fig. 1b). Interestingly, a deficit relative to the modern atmospheric composition has been observed in several paleo-atmospheric samples[21,23,60]. The fact that this deficit appears to vanish around the time of the GOE has been tentatively attributed to an episode of extensive outgassing related to mantle evolution in the late Archean[60]. More data are needed to assess the magnitude and timing of the disappearance of this $^{129}Xe$ deficit[43].

## Box 2 | Missing atmospheric Xe: alternative scenarios?

Previous attempts to explain missing Xe have proposed either a diminished or enhanced reactivity of Xe with deep Earth minerals. For example, the significantly higher solubility of Ar compared to Xe in $MgSiO_3$ perovskite led Shcheka and Keppler[114] to propose that Xe could have been depleted early in the Earth's lower mantle during the crystallization of perovskite from a magma ocean, relative to the lighter noble gases. A Xe-enriched primordial atmosphere could have subsequently escaped to space, leaving later mantle degassing to replenish Ar-Kr, and to a lesser extent Xe, in the atmosphere. As with all the models discussed below, this hypothesis fails to explain the isotopic evolution of atmospheric xenon observed throughout the Archean eon.

Xenon sequestration in a variety of terrestrial reservoirs (Bekaert et al.[42] *and references therein*) and preferential retention in the solid Earth relative to other noble gases during degassing have been proposed to contribute to Xe elemental depletion in the atmosphere. Due to the limited storage capacity of the associated surface reservoirs, the shale, clathrate, and ice hypotheses can be ruled out. However, whilst Xe is relatively inert under ambient and neutral conditions, the potential for its enhanced reactivity at high-temperature and high-pressure[16] and possible incorporation into mineral phases at depth, is commonly used to argue for missing atmospheric Xe to be stored in the Earth's interior. Although theoretical and experimental investigations suggest possible Xe incorporation into silicate phases found in the Earth's crust[46], even the highest measured crustal concentrations of Xe are still three orders of magnitude below that required to account for the missing Xe[115], therefore undoubtedly excluding the upper continental crust as the main "missing" atmospheric reservoir. Likewise, Xe abundance in volcanic rocks and xenoliths indicates that the upper mantle reservoir contains 10–100 times less Xe than the present-day atmospheric Xe inventory[118], making it an unlikely resting place for missing atmospheric Xe and leaving the deeper mantle and/or core as the only possible sinks. Importantly, explaining the underabundance of atmospheric Xe (relative to the noble gas abundance pattern in meteorites) by trapping it in silicate reservoirs without mass-dependent isotopic fractionation could only account for the low amount of xenon in the Earth's atmosphere and not for its strong enrichment in heavy isotopes relative to light ones. Storing the missing xenon inside the Earth over long timescales without re-equilibrating its composition with that of the atmosphere (i.e., via degassing and subduction) presents another major challenge.

Recently, Rzepliński et al.[28] measured the isotopic composition of Xe trapped in natural feldspar and olivine samples confined at high pressures and high temperatures with air- or nitrogen-diluted Xe and Kr. These authors reported mass-dependent isotope fractionation up to +2.3‰ per atomic mass unit, accompanied by strong Xe enrichment

relative to Kr[28]. These findings were interpreted as evidence that Earth's modern Xe isotope signature reflects repeated (between 9 and 15 events) interactions between a primary atmosphere and crystallizing magma oceans[28]. In each episode of isotopic equilibration, heavy Xe isotopes would have been preferentially sequestered in deep silicate phases with elevated Xe/Kr ratios. Atmospheric loss driven by high-energy impacts and hydrodynamic escape would have selectively removed the successive atmospheres enriched in lighter Xe isotopes. A late influx of chondritic material toward the end of Earth's accretion would have then reset the atmosphere to a CI-like composition, which would have then evolved during the Archean through partial remobilization of heavy Xe previously trapped in deep silicate phases[28]. This process is proposed to have produced the observed transition from a CI chondrite-like to a present-day atmosphere, as observed from fluid inclusion analyses[22]. However, this scenario presents significant caveats. For example, how the release of mantle Xe – repeatedly enriched in Xe relative to other noble gases (including Kr) and compared to a CI-like starting composition—could ultimately produce an atmosphere with a Xe deficit (still relative to Kr and CI chondrites) remains to be quantitatively assessed. Rzepliński et al.[28] suggest that the enrichment of light Xe isotopes in mantle gases may be explained by the contribution of recycled Archean atmosphere to the mantle source. However, the idea that the mantle is both (i) the driver of atmospheric evolution through degassing and (ii) the reservoir preserving ancient atmospheric compositions appears problematic. Ultimately, the reason why the Xe isotopic evolution of Earth's atmosphere eventually stopped has yet to be explained in the context of this model.

Storing missing atmospheric Xe in a mantle reservoir would arguably require this reservoir to exhibit a Xe enrichment relative to other noble gases. Recent analyses of heavy noble gases in oceanic island basalts suggested that, rather than displaying an Xe excess, deep plume mantle sources display a singular xenon depletion probably acquired early in Earth's histor[119]. As such, deep plume mantle sources do not contain missing atmospheric Xe. Missing xenon components in the deep Earth and in the atmosphere may represent two distinct problems[119]. While fluid inclusion analysis has now provided a wealth of knowledge that allows one to attribute missing atmospheric Xe to the protracted Xe escape to space throughout the Hadean and Archean (Fig. 1), the cause(s) of missing Xe in the deep mantle remain(s) to be explored. One potential way to advance this problem would be to analyze noble gases in ancient (Archean or older) mantle rocks and minerals (e.g., diamonds), which represents a formidable challenge given the paucity of such samples, potential issues associated with their preservation over eons, and the analytical challenges associated with the analysis of such samples with low gas abundances.

## Major element composition of the ancient atmosphere

Fluid inclusion analyses provide a unique opportunity to probe the major volatile element composition of the ancient atmosphere. However, constraining the partial pressure of ancient atmosphere gases from elemental ratio measurements in fluid inclusions is not trivial due to multi-component mixing (Box 1) and phase chemistry.

### Fluid inclusion phase chemistry

While it has often been assumed that gas compositions measured from fluid inclusions can be directly interpreted as reflecting the atmospheric conditions at the time of mineral precipitation, there is growing evidence that solubility effects associated with the partitioning of volatiles between gas and aqueous phases present at the time of inclusion formation must be considered to avoid misinterpretation[61]. Because each gas has a different solubility in water, the composition of gaseous inclusions (e.g., trapped air

bubbles) can differ markedly from that of the dissolved gases in entrapped fluids. For instance, while the modern atmosphere is composed of about 78.1% $N_2$, 20.9% $O_2$, 0.9% Ar, and ~420 ppm $CO_2$, these proportions shift significantly upon dissolution in freshwater at 20 °C, yielding a composition of about 63.0% $N_2$, 33.4% $O_2$, 1.6% Ar, and 1.9% $CO_2$ (Fig. 3). These proportions depend on the salinity of the fluid phase and temperature of the system, yielding about 70.3% $N_2$, 25.7% $O_2$, 1.8% Ar, and 2.3% $CO_2$ for seawater-like fluids at 90 °C (Fig. 3). As a result, elemental ratios such as $N_2$/Ar may vary from ~38 in fresh air-equilibrated water at 20°C, up to ~84 in pure air bubbles. Park & Schaller[61] emphasized the importance of these considerations and proposed a robust approach using the $N_2$/Ar as a proxy for calculating the gas volume fractions ($\varphi_g$; Fig. 3) at the time of entrapment, therefore allowing the observed gas ratios to be corrected to accurately reflect the composition of the atmosphere under which the fluid inclusions formed. Recent studies have also shown that estimates of the paleo-

## Box 3 | Heavy noble gas systematics in Earth's mantle, crust, and surface through time

Noble gas isotopes come in two distinct isotopic "flavors". The first one corresponds to the primordial isotopes, which are not produced by radioactive processes and have therefore been present on Earth since accretion. These isotopes are crucial to identify the cosmochemical source(s) of terrestrial volatiles. The second "flavor" is the secondary isotopes, which are produced through time by nuclear (such as radioactive decay and spontaneous fission) processes and can therefore provide constraints on the timing of reservoir formation and degassing history. As time passed and mantle reservoirs underwent degassing into the atmosphere, the abundances of primordial noble gas isotopes in the solid Earth decreased. Conversely, as long as a parent nuclides are alive, the production of radiogenic isotopes continuously increases the abundance of these secondary isotopes in the solid Earth (Box Fig. 2). Compared to the atmosphere, Earth's interior is largely enriched in radiogenic products (e.g., $^{40}Ar^*$ from $^{40}K$ decay and $^{129}Xe^*$ from $^{129}I$ decay; Box Fig. 2) relative to primordial isotopes. For Ar, calculations by Bender et al.[15]

suggest that ~75% of the total $^{40}Ar^*$ outgassing from the solid Earth today (~$1.12 \times 10^8$ mol/yr) originates from the continental crust (~$0.85 \times 10^8$ mol/yr), with the remaining ~25% (~$0.27 \times 10^8$ mol/yr) deriving from the mantle. For $^{129}Xe^*$, the formation of the continental crust arguably occurred after $^{129}I$ became extinct (i.e., after ~100 Myr after the formation of the Solar System), implying that virtually 100% of the outgassing $^{129}Xe^*$ originates from the mantle. The progressive degassing of the solid Earth to the atmosphere through time is expected to have resulted in a progressive change of its isotopic composition, which may be recorded in ancient air samples. Such an atmospheric evolution, which has been proposed for Ar[15,57] and Xe[60], is expected to also apply to lighter noble gases. Indeed, mantle reservoirs are largely enriched in primordial $^{20}Ne$ relative to $^{22}Ne$ compared to the atmospheric composition. Ongoing analytical developments hold promises for using these isotopes to better constrain the evolution of Earth's mantle outgassing over time[58].

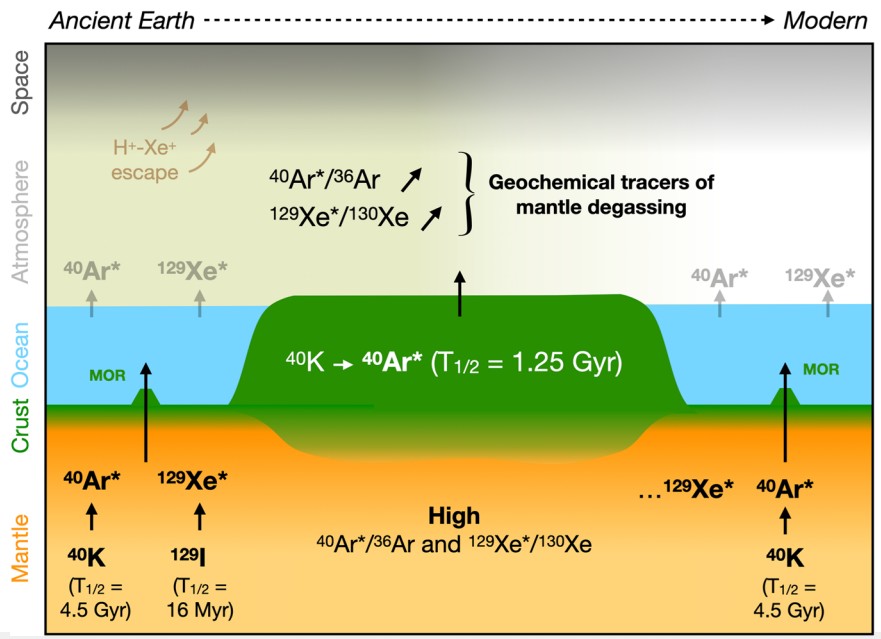

**Box Fig. 2 Production of radiogenic noble gas isotopes in the solid Earth, and progressive transfer to Earth surface environments via degassing.** Radiogenic $^{40}Ar^*$ and $^{129}Xe^*$ were produced in Earth's history by electron capture on extant $^{40}K$ ($T_{1/2}$ = 1.25 Gyr) and β-decay of extinct $^{129}I$ ($T_{1/2}$ = 15.7 Myr), respectively[120]. When compared to primordial Ar ($^{36}Ar$) and Xe ($^{130}Xe$) isotopes, the relative abundances of $^{40}Ar^*$ and $^{129}Xe^*$ in the solid Earth are much greater than in the atmosphere. As a result, outgassing from the continental crust and mantle release high $^{40}Ar^*/^{36}Ar$ and $^{129}Xe^*/^{130}Xe$ gas that can modify the composition of the atmosphere through time[15,60].

atmospheric elemental ratio can be obtained from crushing experiments on ancient hydrothermal quartz and baryte[43,62], although the true Kr/Xe of the Archean atmosphere remains elusive, given the number of processes able to impart elemental fractionation[62].

Documenting the volatile element composition of the ancient atmosphere is key to understanding the environmental, geological, and climate events that accompanied early life evolution. One of the strongest constraints on Archean atmospheric composition is that the ground-level mixing ratio of $O_2$ was <$10^{-6}$ PAL (present atmospheric level; Fig. 1), and that the release of $O_2$ by early cyanobacteria as a byproduct of oxygenic photosynthesis dramatically changed Earth's atmosphere, transforming Earth's weakly reducing, anoxic atmosphere into an oxygenated one during the GOE, ~2.5 Gyr ago. Estimates of other Archean atmospheric gas concentrations

are subject to significant uncertainty, with for example $CO_2$ and $CH_4$ levels ranging ~10 to 2500 and $10^2$ to $10^4$ times modern amounts, respectively[29]. Interestingly, there exists an apparent contradiction between astrophysical models, which suggest that the Sun's luminosity was about 25–30% weaker during the Archean eon and so the Archean Earth should have been frozen, and geological evidence indicating that Earth's surface temperatures were warm enough to support liquid water and early life[63]. This so-called faint young sun paradox is thought to be resolved by higher concentrations of greenhouse gases (such as $CO_2$, $CH_4$, $H_2$) in the early atmosphere[64].

Fluid inclusions trapped in evaporites (e.g., halite) also provide an opportunity to probe the gas composition of the ancient atmosphere, up to the Neoproterozoic period[65]. Over the past decades, great strides have been achieved in advancing techniques of gas extraction from halite fluid

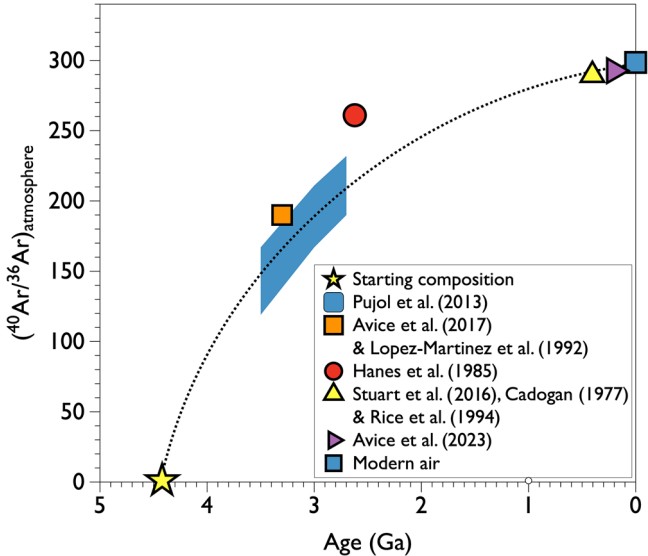

**Fig. 2 | Evolution of the isotopic composition of atmospheric argon over time.**
The progressive increase of the atmospheric $^{40}Ar/^{36}Ar$ ratio is attributed to the
outgassing of radiogenic $^{40}Ar$ from both the Earth's mantle and crust[15]. The dashed
line represents a recent model published by Zhang et al.[58] for the evolution of
atmospheric argon. Error bars (1 sigma) are contained within the symbols[19,21,109–113].
For Pujol et al.[57], a possible range of values is indicated in orange. The starting
composition ($^{40}Ar/^{36}Ar$~0) is hypothetical and reflects the original contribution of
primordial argon (primarily $^{36}Ar$) from chondritic bodies. This inference is also
supported by the chondritic-like $^{38}Ar/^{36}Ar$ ratio of the Earth's atmosphere.

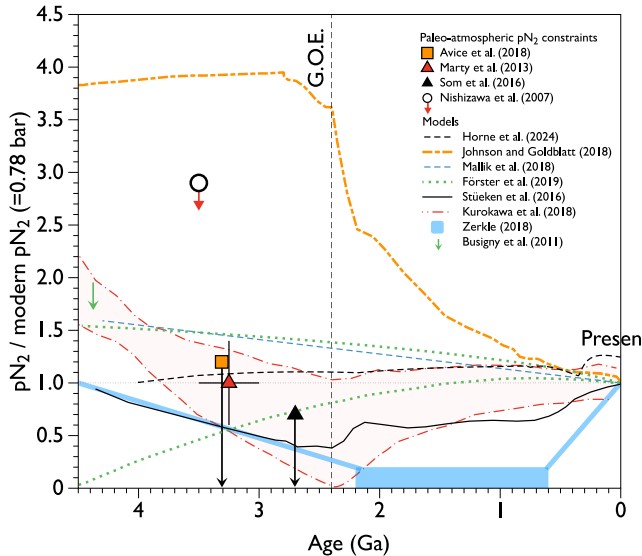

**Fig. 4 | Compilation of models and estimates of the evolution of the partial
pressure of dinitrogen in the Earth's atmosphere.** For the Archean, available data
suggest that the $pN_2$ was on the same order of magnitude as or lower than the
modern value. However, all models predict fluctuations of more than 50% over
Earth's history. Dramatic changes in the $pN_2$ could have occurred after the Great
Oxidation Event[75], but the lack of data does not allow one to reliably test this
hypothesis[22,68,70–75,77,114,115].

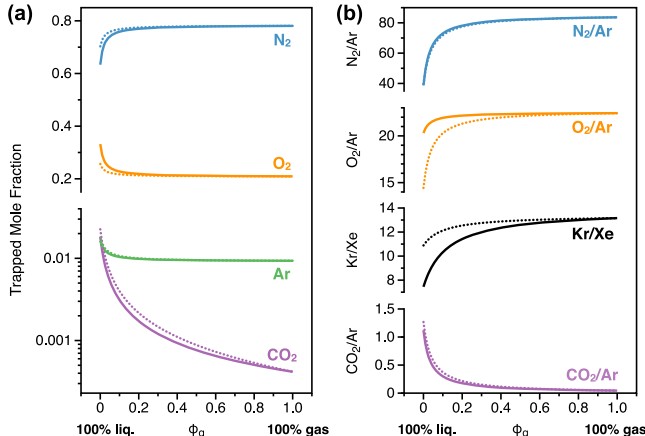

**Fig. 3 | The gas composition of fluid inclusions is modeled as a mixture between
modern atmospheric air and air-saturated water (modified from Park & Schaller
(2025)[61]).** The resulting composition depends on the relative contribution of each
endmember, governed by the gas volume fraction ($\varphi_g$). When $\varphi_g = 1$, the inclusion
reflects the atmospheric composition, when $\varphi_g = 0$, it reflects the composition of
dissolved gases in water. Theoretical trapped mole fractions of $N_2$, $O_2$, $Ar$, and $CO_2$
are shown on (**a**), with the wide range of possible $CO_2$ mole fractions shown alto-
gether with $Ar$ on a logarithmic $y$-axis. Ranges of selected elemental ratios are shown
on (**b**). All calculations using solid lines assume 20 °C freshwater, whereas dotted
lines show calculations for 90 °C seawater, for comparison.

## Evolution of atmospheric N2 through time

Today, $N_2$ is the most abundant gas in the atmosphere, reflecting the long-
term evolution of Earth's surface as a result of biological and geological
processes, including mantle degassing (e.g., volcanism) and ingassing (e.g.,
via subduction)[13]. A key question is whether the partial pressure of atmo-
spheric $N_2$ has evolved over billion-year timescales due to the complex
interplay between processes contributing to the geological nitrogen cycle.
Nitrogen primarily accumulates either as atmospheric $N_2$ or within rocks in
the form of ammonium, amide, nitride, or organic nitrogen. Under typical
mantle temperatures and redox conditions, volcanic gases release $N_2$, which
is chemically inert and therefore eventually enters the atmosphere. How-
ever, the current debate centers on whether the terrestrial nitrogen cycle is in
a net ingassing or degassing regime, primarily because of significant
uncertainties regarding the efficiency of nitrogen recycling into the mantle
via subduction[13,67,68].

Throughout the Phanerozoic, sedimentary C/N data suggest that the
release of $N_2$ into the atmosphere was largely offset by nitrogen burial in
organic matter, leading to minimal fluctuations in the partial pressure of
nitrogen, $pN_2$[69]. Most models of $pN_2$ evolution in the deeper past[70–75] predict
significant changes ( ± 50% of the modern value) over the 4.5 billion years of
Earth's history (Fig. 4). For the GOE and the Proterozoic, some models even
suggest a near collapse of $pN_2$[74], with a potential late recovery around 0.6
billion years ago[75]. However, only a few reliable estimates of the paleo-$pN_2$
are available in the literature (Fig. 4). The nitrogen isotope composition of
fluid inclusions in ancient rocks indicates that the $^{15}N/^{14}N$ ratio of atmo-
spheric nitrogen at 3.3 Ga was already modern-like[22,76], attesting to the
inefficient escape of nitrogen via fractionating escape mechanisms since that
time. Note that potential isotopic effects related to phase chemistry (i.e.,
difference in solubility between $^{15}N$ and $^{14}N$) are minor, lower than the part
per thousand (permil) level[61].

As of today, analyses of nitrogen and argon (i.e., $N_2/^{36}Ar$) in fluid
inclusions from Archean hydrothermal minerals suggest that the paleo-$pN_2$
in the ancient atmosphere was similar to or lower than that of the present
atmosphere at 3.3 Ga, and <1.1 bar at 3.5 to 3.0 Ga[22,76,77]; Fig. 5). In detail,
Marty et al.[76] proposed that the $pN_2$ during the Archean was similar to the
modern value. In a subsequent study, using new samples that exhibited well-

inclusions (e.g., heat or cold extraction followed by mass spectrometry
analyses), making it possible to increase resolution while reducing sample
size requirements[66]. Critically, however, conservative tracers of gas-fluid
partitioning (e.g., $^{40}Ar/N_2$[61], Fig. 3) are invariably required to provide robust
insights into the ancient atmosphere's composition.

defined correlations in the $^{40}Ar/^{36}Ar$ vs. $N_2/^{36}Ar$ space, Avice et al.[22] confirmed that Archean $pN_2$ was arguably not higher than the modern value and further suggested it may have been lower. These data do not put any stringent constraint on how low the $pN_2$ could have been during the Archean eon. There exist lingering uncertainties about whether a direct link can be established between the measured $N_2/^{36}Ar$ ratio of the gas released from fluid inclusions and the "true" atmospheric $N_2/^{36}Ar$ ratio[22,76], and additional work is therefore needed to better constrain the exact $pN_2$ of the Archean atmosphere. Due to the ubiquitous presence of a hydrothermal component rich in crustal $N_2$ (high $N_2/^{36}Ar$) and radiogenic $^{40}Ar^*$ (high $^{40}Ar/^{36}Ar$[77]), existing measurements often face difficulties in clearly determining the composition of the paleo-atmospheric endmember and the value of paleo-atmospheric $pN_2$ (Fig. 5). For the Marty et al.[76] dataset, for example, the analyzed fluid inclusions arguably represent hydrothermal fluids derived from seawate[78,79]. While the occurrence of air bubbles can be discarded based on the intra-

ocean origin of the host minerals, a contribution from magmatic $CO_2$ cannot be eliminated. Based on mass balance calculation, however, the $N_2/Ar$ ratio measured from fluid inclusion would still represent that of air-equilibrated water rather than a magmatic component. Future measurements on samples with a higher relative proportion of the paleo-atmospheric component[19] might help determine the $pN_2$ of the ancient atmosphere with greater precision. Analysis of clumped $N_2$[80] from ancient fluid inclusions represents an analytical challenge that would offer a promising avenue for distinguishing atmospheric-derived and hydrothermal nitrogen.

While $N_2$ and Ar arguably have similar solubilities in basaltic melts (implying no significant fractionation of the $N_2/^{36}Ar$ during mantle degassing, at least at the current mantle oxidation state[81]), the atmospheric $N_2/^{36}Ar$ could have evolved through time due to the distinct recycling efficiencies of these elements during subduction. The recycling efficiencies of both elements during early subduction processes associated with Hadean and Paleoarchean plate tectonics[82,83] would arguably have been low due to the high mantle temperatures (Fig. 1b). The emergence of colder, modern-style subduction (whereby a significant vertical volatile flux of surface materials to mantle depths) would have marked a pivotal shift in surface-mantle interactions[84], transitioning from a purely degassing state to a balance between global degassing and ingassing (i.e., recycling[13]). While recycled Ar is primarily hosted in hydrated phases like serpentinites (and thus potentially lost to the mantle wedge during dehydration[85]), N can be retained in metasediments and peridotites due to $NH_4$ substitution for K in phengite or $N_3^-$ substitution for oxygen[86,87], implying no significant loss of N by dehydration[68,88]. The recycling efficiencies of Ar and N into the mantle via subduction may thus differ significantly due to their distinct retention behaviors in subducted materials[13]. While heavy noble gas (Ar, Kr, Xe) systematics clearly suggest that substantial recycling of surface volatiles into the mantle has occurred over the past ~2–3 billion years[89,90], the global recycling efficiency of nitrogen remains open to debate, and the potential for past atmospheric $N_2/^{36}Ar$ variations remains uncertain.

## Evolution of the oceans' salinity

Determining the salinity of ancient oceans is a key objective in geosciences, as salinity significantly influences Earth's climate[91] and provides insights into the conditions under which life first emerged[92] (Box 4). Kasting[93] proposed that ancient ocean salinity might have been approximately 1.2 times higher than present levels, assuming that all evaporites currently located on continents were once dissolved in the oceans. However, the full extent of Precambrian evaporites remains uncertain due to poor preservation. Building on this idea, Knauth[92] considered the role of brines in stratified oceans and those potentially preserved on continents, suggesting that Archean Ocean salinity may have ranged from 1.2 to 2 times that of today. Knauth[92] identified evaporite formation and its isolation on continental platforms as the primary processes for decreasing ocean salinity, positing that most halogens likely remained dissolved in the oceans until major

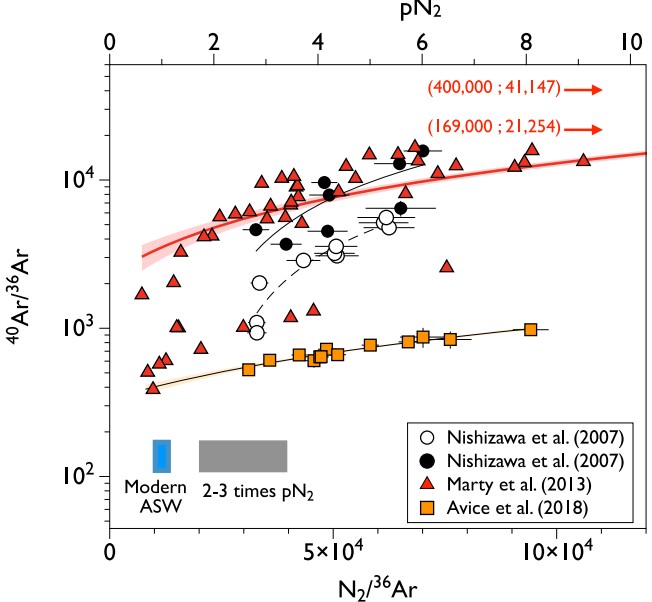

**Fig. 5 | Mixing correlations between paleo-atmospheric gases and crustal/hydrothermal gases contained in fluid inclusions of Archean (3.5–3 Ga) hydrothermal quartz from various geological settings.** Note the logarithmic scale on the y-axis. The atmospheric endmember has an estimated $^{40}Ar/^{36}Ar$ ratio of $140 \pm 30$[57] (Fig. 2). Uncertainties are shown at 1 sigma. Two points from Marty et al.[76] plot outside of the displayed abscise range and are therefore given in parenthesis. These data suggest that the partial pressure of nitrogen ($pN_2$) in the ancient (including Archean) atmosphere was arguably not 2–3 times greater than present-day.

## Box 4 | Temperature–salinity of the Archean oceans: implications for early life

The characteristics of the Archean oceans, including salinity, temperature, and chemical composition, were essential factors in the development of early life. Higher salinity levels compared to modern seawater would have contributed to warming up the planet[91], but could also have hindered the evolution of macroscopic marine organisms during the Archean eon[121]. The growth of larger life forms might have been restricted since metazoans are intolerant of high salinity environments[122], which aligns with the prevalence of cyanobacteria (relatively salt-tolerant) in the Precambrian fossil record[92]. However, fluid inclusion data suggest that the salinity of the Archean oceans might have been similar to that of modern oceans, implying that early Archean life was not confined to salt-tolerant species or more dilute

waters like estuaries[122] and could have been widespread across the oceans. This also raises the possibility that significant amounts of oxygen were present in the Archean oceans, as oxygen solubility in seawater increases with decreasing salinity and temperature[123]. Despite the assumption of relatively stable global ocean salinity over time, localized saltier environments during the Archean are still possible, as suggested by the presence of highly saline ancient seawater trapped in fluid inclusions within 3.2-Ga quartz crystals[116]. Additionally, the presence of evaporites in the Archean geological record and evidence of saline environments point to the potential for localized evaporation, forming habitats like lagoons that could have supported life[124].

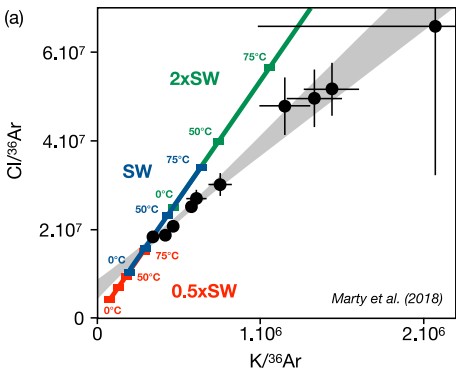

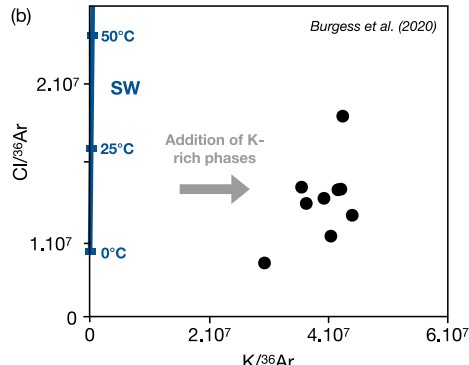

**Fig. 6 | Geochemical approach to reconstructing ancient ocean salinity.** Mixing correlations between a paleo-ocean and a crustal/hydrothermal endmember in fluid inclusions of **a** 3.49-Gyr-old Dresser formation, Warrawoona Group, Pilbara Craton at North Pole (Western Australia[96]), and **b** 2.5 Gyr-old chert samples from Hamersley Group (Western Australia[97]). **a** The strong linear correlation indicates two-component mixing. The red, blue, and green thick lines represent salinities of 0.5, 1, and 2 times the modern seawater composition, as expressed by Cl and K, for temperatures of 0 °C, 25 °C, 50 °C, and 75 °C. These ranges represent possible endmembers between present-day ocean bottom temperatures of 2 °C and those proposed for Archean oceans of up to 70 °C[99]. The correlation is not consistent with salinities twice the modern value[116] or higher[55,56,117]. For a modern-like salinity, a temperature in the range of 20–40 °C is compatible with the data. **b** Only the main release steps (at 1400 or 1600 °C) from each sample are displayed here. They show Cl/$^{36}$Ar values comparable to modern seawater salinity in the temperatures range 0–25 °C, but K/$^{36}$Ar about two orders of magnitude higher than modern seawater, which the authors attribute to the presence of a K-rich phase[97]. Based on the low $^{40}$Ar/$^{36}$Ar, seawater-like Cl/$^{36}$Ar, and the ~2.4 Ga $^{40}$Ar–$^{39}$Ar ages, the fluids are likely to be paleo-seawater containing dissolved atmospheric noble gases.

episodes of continental growth. Yet, a further mechanism for halogen removal is subduction and recycling into the mantle, as recent studies on halogen budgets in continental arc environments suggest (ref. 94 *and references therein*). The fluxes of halogens between the mantle, oceans, and oceanic crust are still not well understood[95], making it difficult to estimate ancient ocean salinity based solely on geochemical mass balance.

Available data from fluid inclusions in Archean minerals[96,97] suggest that the salinity of Archean oceans was comparable to that of modern oceans within a plausible temperature range of 0–75 °C (Fig. 6). Many geochemical proxies have been used to document the salinity and temperature of the Archean Oceans. Oxygen and silicon isotope variations in ancient cherts and organic matter suggest that Archean seawater temperatures were likely greater than modern oceans, with estimates ranging from around 55 °C to 85 °C[98–100]. Other studies have argued for more moderate ocean temperatures around 40 °C, suggesting that early life may have thrived in cooler, more temperate conditions[101]. In any case, fluid inclusion data appear most compatible with a modern-like salinity of the Archean oceans, compatible with conclusions from mass balance of hydrothermal activity and weathering rates[10]. Specifically, although the similar Br/Cl and I/Cl ratios suggest no significant changes in the ocean's halide system between 2.5 and 3.5 billion years ago compared to modern seawater, Burgess et al.[97] suggested that the ancient ocean had higher levels of bromine (Br) and iodine (I) relative to chlorine (Cl). Because iodine exhibits a strong affinity for organic matter, higher iodine concentrations in the Archean Ocean compared to today could potentially reflect reduced biological sequestration, assuming the total organic reservoir was smaller than at present. This scenario may be consistent with the elevated Br/Cl ratios observed in Archean seawater[102], although the variable influence of mantle-derived hydrothermal vent inputs cannot be ruled out[97].

## Conclusions and perspectives

- Fluid inclusions represent invaluable time capsules that preserve geochemical information about the environmental conditions during hydrothermal mineral precipitation.
- The original geochemical signatures of fluid inclusions may be overprinted by later events (e.g., hydrothermal circulation associated with regional metamorphism), thus calling for caution when reconstructing paleoenvironmental conditions from fluid inclusion analyses[103].

- Noble gas isotopes provide a complex yet comprehensive toolset to track both the evolution of the mantle (e.g., $^{129}$Xe deficit, evolution of atmospheric $^{40}$Ar/$^{36}$Ar) and the atmosphere (e.g., time evolution of Xe isotopes via mass-dependent fractionation) through time.
- Coupling Xe isotope analyses of ancient geological materials with models of atmospheric Xe photochemistry shall help resolve the missing Xe paradox, provided that a physical process capable of transporting Xe through the atmosphere is identified.
- Xenon isotopes are key to tracking the evolution of hydrogen escape, with key implications for our understanding of solar activity, terrestrial magnetic field, and evolution of redox conditions at Earth's surface (including ocean pH). The possibility that this evolution was discontinuous[24] presents an intriguing opportunity to potentially reconstruct the evolution of solar activity and/or the terrestrial magnetic field over time.
- The evolution of $pN_2$ provides indirect constraints on past $pCO_2$, which has significant implications for understanding major scientific questions such as the faint young sun paradox[104] and the evolution of the biosphere through geological time.
- The evolution (or lack of evolution) of ocean salinity through time has implications for the emergence of continents, the flourishing of life, and the global balance (ingassing vs. degassing) of volatile elements via subduction/volcanism.
- Based on fluid inclusion data, the salinity and partial pressure of $N_2$ in the atmosphere were comparable to present-day levels during the Archean eon. These findings imply that the relative fluxes of $H_2O$, Cl, and $N_2$ to and from Earth's surface reservoirs (i.e., oceans and atmosphere) have remained relatively constant since the Archean.
- Geochemical constraints from fluid inclusion data can be considered alongside other geochemical proxies of past environmental conditions (e.g., Si isotopes in cherts[99]) to provide a holistic representation of early Earth. Additionally, these geochemical data must be compared with model outputs (e.g., regarding the evolution of $pN_2$) to improve our understanding of the mechanisms controlling the evolution of Earth's surface environments.
- The exact composition (abundance and isotope signature) of nitrogen in the ancient atmosphere remains uncertain. Additional, high precision analyses are required. One promising avenue of investigation may be to explore the nitrogen clumped isotope composition[80] of ancient atmospheric samples trapped in FIs.

- Novel crushing techniques for fluid inclusion extraction[105,106], combined with ongoing analytical developments in nitrogen and noble gas mass spectrometry[107]—including the emerging possibility of measuring all noble gases, and potentially nitrogen, in the same gas fractions[19,76,77,108] — hold great promise for advancing the use of fluid inclusions as tiny windows into Earth's past environments.
- Parallel studies of the origin and evolution of atmospheres of other terrestrial planets (Venus and Mars) and remote analyses of exoplanetary atmospheres offer insights into the emergence and development of habitable conditions on other worlds.

## Data availability

No new datasets were generated in this study. All data used are available in the cited literature.

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

## Acknowledgements

D.V.B. acknowledges funding from the Agence Nationale de la Recherche (Grant ANR-22-CPJ2-0005-01). G.A. has received funding from the European Research Council (ERC) under the European Union's Horizon Europe research and innovation program (Project ATTRACTE, grant agreement no. 101041122). B.M. also acknowledges funding from the European Union (ERC: PHOTONIS, grant 695618).

## Author contributions

D.V.B. wrote the original draft of this manuscript, which was then edited by G.A. and B.M. before submission.

## Competing interests

The authors declare no competing interests.
