## [Peer review file · Communications Earth & Environment]

Fluid inclusions: tiny windows into global paleo-environments

Corresponding Author: Dr David BEKAERT

Version 0:

Decision Letter:

Dear Dr BEKAERT,

We apologize for the delay in sending our decision letter.

Your manuscript titled "Fluid inclusions: tiny windows into global paleo-environments" has now been seen by 3 reviewers, whose comments are appended below. You will see that they find your work of some potential interest. However, they have raised quite substantial concerns that must be addressed. In light of these comments, we cannot accept the manuscript for publication, but would be interested in considering a revised version that fully addresses these serious concerns.

We hope you will find the reviewers' comments useful as you decide how to proceed. Should additional work allow you to address these criticisms, we would be happy to look at a substantially revised manuscript. If you choose to take up this option, please either highlight all changes in the manuscript text file, or provide a list of the changes to the manuscript with your responses to the reviewers.

In particular, please ensure that the revised manuscript meets the following editorial requests:

A restructuring of the manuscript is necessary to expand the conceptual framework of the research and broaden the discussions by citing relevant and recently published articles.

Specifically, the approach and considerations on noble gases should be compared to similar approaches, and the scientific debate on relevant topics should be acknowledged and discussed in depth. Additionally, the state-of-the-art analytical developments in the field should be expanded, and recent literature that informs the understanding of the ancient atmosphere using different geochemical tracers should be incorporated, as recommended by the reviewers.

When resubmitting, please provide a point-by-point response to the reviewers' comments. Please submit your responses as a separate file, distinct from your cover letter where you can add responses to the Editors' comments that you do not want to be made available to the reviewers. Word files are preferred. We recommend that any figures, tables or graphs that are included in the response to reviewers are also included in the main article or Supplementary Information.

If the revision process takes significantly longer than three months, we will be happy to reconsider your paper at a later date, as long as nothing similar has been accepted for publication at Communications Earth & Environment or published elsewhere in the meantime.

We are committed to providing a fair and constructive peer-review process. Please do not hesitate to contact us if you wish

to discuss the revision in more detail.

Please use the following link to submit your revised manuscript, point-by-point response to the reviewers' comments with a list of your changes to the manuscript text (which should be in a separate document to any cover letter), a tracked-changes version of the manuscript (as a PDF file) and any completed checklist:

Link Redacted

Please do not hesitate to contact us if you have any questions or would like to discuss the required revisions further. Thank you for the opportunity to review your work.

Best regards,

Carolina Ortiz Guerrero, Ph.D.
Associate Editor
Communications Earth & Environment

EDITORIAL POLICIES AND FORMAT

If you decide to resubmit your paper, please ensure that your manuscript complies with our editorial policies and complete and upload the checklist below as a Related Manuscript file type with the revised article:

Editorial Policy Policy requirements
(Download the link to your computer as a PDF.)

- Behavioural and social science
- Ecological, evolutionary & environmental sciences
- Life sciences

<https://www.nature.com/documents/nr-reporting-summary.zip>

For your information, you can find some guidance regarding format requirements summarized on the following checklist: (<https://www.nature.com/documents/commsj-phys-style-formatting-checklist-article.pdf>) and formatting guide (<https://www.nature.com/documents/commsj-phys-style-formatting-guide-accept.pdf>).

REVIEWER COMMENTS:

Reviewer #1 (Remarks to the Author):

This article aims at reviewing what has been learnt from hydrothermal fluid inclusions on the evolution of paleo-environments, i.e. paleo-atmosphere and paleo-oceans. Indeed, much has been done and this topic is of interest to a large community in Earth&planetary sciences. The manuscript however mostly covers previous findings by the authors, and is unbalanced in two ways: 1) nearly half of it is focused on the missing Xe paradox and the authors own scenario, 2) some references are missing or misleadingly cited, others are rejected without discussion or based on arguments that the reader cannot evaluate, being given only a very one sided view. This is quite far from a healthy scientific debate, as expected in a review paper. The article also contains 4 boxes to give a pedagogic focus on some items, but 3 of them are oversimplified and miss key aspects.

For these reasons, I cannot recommend publication of this article.

More detailed comments:

1- A minimum of information on fluid inclusions in quartz (or other minerals) should be given, their geological context, type of host rocks, proper description of fluid inclusions distribution. For many readers, the title might be misleading as fluid inclusions are widely studied in magmatic/volcanic contexts. This article targets one specific type of fluid inclusions, which is fine of course but should be stated somewhere to avoid confusions.

2- Box 1: 'two populations of fluid inclusions, i.e. two different sizes, would result from two different events'. Such a description of fluid inclusions does not correspond to the reality and wrongly gives the impression that simple crushing steps can separate populations.

Fluid inclusions in hydrothermal quartz grains are typically less than few tens of μm , and a single mineral grain contains

several generations of fluid inclusions with distinctly different compositions

3- p.116: 'over geological time, the composition of the atmosphere results from a balance between volcanic outgassing and ingassing of the solid Earth via subduction (Bekaert et al. 2021)'. This is mostly true for CO₂, but not for other volatiles, noble gases included, for which volcanoes are just one way to outgas solid Earth, erosion and metamorphism are other essential processes (see Bender et al. 2008 for Ar for instance). This requires much more discussion as it is quite key to the formation of fluid inclusions in minerals, and the discussion should be supported by more references than just one by the author.

4- I.132: 'Noble gases are inert'. Further down the authors cite Grochala 2007, a review paper on noble gases compounds. It should be specified here that noble gases (xenon mostly) may not be inert at all conditions. The literature on the topic is quite extended, with several reviews more recent than Grochala 2007.

5- I.151. 'Unlike lighter krypton, xenon is easily ionized by solar UV or charge exchange with H⁺ ions, so Xe⁺ can be dragged out to space by escaping H⁺ ions without significantly affecting atmospheric Kr.' This statement is not that easy to follow since Xe ionisation energy (12.13 eV) is only slightly lower than that of Kr (14 eV). Why a small difference would imply such a Xe-specific depletion?

6- I.153 'Ionized Xe could then have then been being dragged along open magnetic field lines and lost to space via ionic coupling with escaping H⁺ (Zahnle et al., 2019) or becoming trapped in organic hazes formed within the CH₄-rich early atmosphere (Hébrard & Marty, 2014).'

Concerning the first hypothesis, the authors must mention that Zahnle et al. do point out that so far, no mechanism to lift Xe (up to high altitudes at which it could ionise) has been found. This is the most problematic aspect of the Xe-escape scenario that must be clearly spelled out.

Concerning the second hypothesis, the existence of such organic hazes is possible but is speculative.

7- As pointed out by Dauphas, not cited here, whatever process lead to Xe fractionation and depletion affected Mars and Earth similarly, and hence cannot be related to mass/gravity as it differs a lot between Mars and Earth.

8- I.227: 'noble gases lighter than Xe did not experience significant escape to space (Ozima & Podosek 1999; Pepin 2006).' Ozima and Podosek did not report that, they reported that only Xe was missing, not that only Xe was lost to space.

9- I.231: No references are given for the thermal escape process (Jeans escape).

10- Box 2.

I.263: Shcheka & Keppler 2012 do not report Xe potential for enhanced reactivity but a high retention of Ar in lower mantle bridgmanite (still named perovskite at the time), unlike for Xe.

I.269: Sanloup et al. 2005 investigated the retention of Xe in quartz.

The scenario by Rzeplinski et al. answers most if not all questions raised in the first half of the manuscript, i.e. elemental and isotopic Xe depletion during the Hadean, and later evolution of the atmosphere by degassing of heavier Xe isotopes initially trapped in the crust/mantle rocks with a systematic preferential release of lighter noble gases (hence of Kr) throughout the Archean. They also provide a thorough analysis of Xe enrichment in the natural rock record (and not just in one instance as done here on I.268) and to which extent it reflects the actual xenon content in rocks at depth due to retrodiffusion upon rock ascent as for any other volatile element. In other words, this scenario cannot be dismissed so lightly but should be properly discussed.

11- I.280 'most of their analyses revealed isotopic variations that do not follow expectations from mass-dependent fractionation'. What kind of isotopic variations are the authors mentioning? This is a very unfair accusation that the readers cannot assess, and considering that reviewers of Rzeplinski et al. 2022 most likely evaluated the reliability of the measurements.

12- Box 3

The authors cite only one reference to support that outgassing of radiogenic ⁴⁰Ar into the atmosphere played a prominent role, while the role of the continental crust remained minor. This contrasts with the results from Bender et al. 2008 that point out the strong contribution from continents, Bender et al. is cited here but misleadingly (in particular in Fig.4 caption, box 3). As pointed out above, the review should be broader, discuss pros and cons of previous results, especially when they differ.

13- L.323: about ¹²⁹Xe deficit observed only in pre GOE atmosphere inclusions, it can indeed be explained by extensive outgassing in the late Archean (Marty et al. 2019), or it can be explained, as the overall Xe isotopic evolution, by the equilibrium reached between Earth's degassing (mantle and crust) and Xe ingassing at the end of the Archean.

14- L.394: 'attesting to the inefficient escape of nitrogen via fractionating escape mechanisms since that time'. How do the results of Marty et al. 2013 and Avicé et al. 2018 compare with atmospheric escape models?

15- The whole literature on volatiles recycling at depth through incorporation in minerals in subduction contexts is ignored (N₂ and Ar in particular).

16- Section 4: it should be reminded here that the original signal might have been erased by later events; the high potential of misinterpreting Archean hydrothermal systems from the investigation of fluid inclusions has been pointed out (Farber et al. 2015 for instance).

Reviewer #2 (Remarks to the Author):

Please see attached. Paper does an excellent job on the noble gas part, but needs a bit more updated literature on the great data available using the bulk volatiles once they are re-interpreted more quantitatively.

Reviewer #3 (Remarks to the Author): [See attached]

Communications Earth & Environment is committed to improving transparency in authorship. As part of our efforts in this direction, we are now requesting that all authors identified as 'corresponding author' create and link their Open Researcher and Contributor Identifier (ORCID) with their account on the Manuscript Tracking System prior to acceptance. ORCID helps the scientific community achieve unambiguous attribution of all scholarly contributions. You can create and link your ORCID from the home page of the Manuscript Tracking System by clicking on 'Modify my Springer Nature account' and following the instructions in the link below. Please also inform all co-authors that they can add their ORCIDs to their accounts and that they must do so prior to acceptance.

Version 1:

Decision Letter:

Dear Dr BEKAERT,

Your revised manuscript titled "Fluid inclusions: tiny windows into global paleo-environments" has now been seen by 3 reviewers, whose comments are appended below. You will see that the reviewers appreciate the effort you put in the revisions, but reviewer 1 continues to raise important concerns about factual mistakes and misconceptions. Given that these issues have been raised before and not fully addressed, we are not certain whether you are able to fully address the referee's concerns; if not, unfortunately we cannot consider your manuscript further, and would therefore recommend you seek publication elsewhere.

In light of these ongoing concerns, although we cannot accept the manuscript for publication, we would be interested in considering a revised version that fully addresses these serious concerns. Specifically, for publication in Communications Earth & Environment to be appropriate, a revised manuscript must include a comprehensive comparative analysis of the literature on the missing Xe paradox, compellingly address atmospheric escape models, and fully incorporate the impact of deep Earth mineralogy on noble gas retention.

We hope you will find the reviewers' comments useful as you decide how to proceed. If additional work allows you to either incorporate or refute these criticisms, we will be happy to look at a substantially revised manuscript. If you choose to take up this option, please either highlight all changes in the manuscript text file, or provide a list of the changes to the manuscript with your responses to the reviewers.

When resubmitting, please provide a point-by-point response to the reviewers' comments. Please submit your responses as a separate file, distinct from your cover letter where you can add responses to the Editors' comments that you do not want to be made available to the reviewers. Word files are preferred. We recommend that any figures, tables or graphs that are included in the response to reviewers are also included in the main article or Supplementary Information.

If the revision process takes significantly longer than three months, we will be happy to reconsider your paper at a later date, as long as nothing similar has been accepted for publication at Communications Earth & Environment or published elsewhere in the meantime.

We are committed to providing a fair and constructive peer-review process. Please do not hesitate to contact us if you wish

to discuss the revision in more detail.

Please use the following link to submit your revised manuscript, point-by-point response to the reviewers' comments with a list of your changes to the manuscript text (which should be in a separate document to any cover letter), a tracked-changes version of the manuscript (as a PDF file) and any completed checklist:

Link Redacted

Please do not hesitate to contact us if you have any questions or would like to discuss the required revisions further. Thank you for the opportunity to review your work.

Best regards,

Alireza Bahadori, PhD
Associate Editor
Communications Earth & Environment
Consulting Editor
Communications Sustainability

EDITORIAL POLICIES AND FORMAT

If you decide to resubmit your paper, please ensure that your manuscript complies with our editorial policies and complete and upload the checklist below as a Related Manuscript file type with the revised article:

- Behavioural and social science
- Ecological, evolutionary & environmental sciences
- Life sciences

For your information, you can find some guidance regarding format requirements summarized on the following checklist: (<https://www.nature.com/documents/commsj-phys-style-formatting-checklist-article.pdf>) and formatting guide (<https://www.nature.com/documents/commsj-phys-style-formatting-guide-accept.pdf>).

REVIEWER COMMENTS:

Reviewer #1 (Remarks to the Author):

The revised manuscript has been corrected for many of the miscitations that were present in the first version of the manuscript, and is now less focussed on Xe, including a broader discussion of atmospheric gases. However, concerning Rzeplinski et al. 2022 whose criticism is quite central in the manuscript, while their scenario is more explicitly laid down now, there are still factual mistakes, misconceptions (see detailed comments below), and most concerning, the disturbing selection of only two analyses out of a whole data-set to build up their criticism. Some key references on the missing Xe paradox have not been added. Overall, the paper lacks a comparative analysis of the literature on the missing Xe paradox and its key characteristics: is there any experimental confirmation attesting Xe isotopic fractionation for each scenario (e.g. atmospheric escape models, accretion models, trapping at depth models)? Same question for Xe vs Kr elemental fractionation? Therefore, despite the improvements, I still cannot recommend it for publication at this stage.

1) I.194: 'Because Xe has a low ionization potential relative to the other noble gases': As mentioned in the previous round of review, this needs a bit of context or quantification, please specify Xe and Kr ionisation potentials. This can alternatively be added later, on I. 311 along with the ionization potential of Ar.

2) The comparison between Mars and Earth is now more deeply described as far as Xe isotopes are concerned. For Xe elemental depletion however, the authors mention Zahnle et al. 2019 who pointed out that the extent of Martian atmosphere Xe depletion is 50% that of the Earth's atmosphere. However, this 50% variation is to be put into context, with atmospheric Xe being elementally depleted by a factor of 24 relative to Kr in CI chondrite. I still do not understand why none of Dauphas's papers on the topic is cited, or his more broader 2014 article (Dauphas and Morbidelli, 2014).

3) I.323: This is supposed to be a review paper. Atmospheric escape models and models considering the effect of deep Earth mineralogy on noble gases retention should be discussed equally as mentioned in the first paragraph above, not

siding blindly on the authors preferred scenario. Box 2 title should be objective, for instance 'Missing atmospheric Xe: alternative scenarios' with or without a question mark.

4) L.357: Rzeplinski et al. 2022 used natural feldspars and olivines, not synthetic ones. Please correct.

5) L. 376-389: 'These findings were interpreted as evidence that Earth's modern Xe signature reflects repeated (between 9 and 15 events) interactions between a reduced primary atmosphere and oxidizing, crystallizing magma oceans.' The mention of a reduced primary atmosphere and oxidizing magma oceans does not correspond to what is described in Rzeplinski et al. 2022. There is no mention of a net gain nor loss of oxygen in their scenario, just Xe substitution to Si in crystals at depth resulting in Xe oxidizing, not the whole magma ocean. The words 'reduced' and 'oxidizing' should be removed.

6) L. 376-389: 'However, this complex scenario suffers from several problems and caveats. While experimental results showed marked deviations relative to the starting isotopic composition, observed isotopic variations did not necessarily follow expectations from mass-dependent fractionation.' There is some improvement here compared to the first version of the manuscript, with a move from 'that most of their analyses revealed isotopic variations that do not follow expectations from mass-dependent fractionation' to 'did not necessarily follow expectations from mass-dependent fractionation. Nonetheless, as the authors claim to be objective in their answer to raised comments, the word 'necessarily' should be quantified. Do the majority of the analyses lie off a mass-dependent fractionation line or only selected outliers? The figure shown for the sake of the review process only, misleadingly leads the reviewers to think it is the original figure from Rzeplinski et al., which it is not (see figure in the pdf file). After careful inspection, it appears that the authors have chosen to reproduce olivine analysis O-01b (as for the original Fig.1, Rzeplinski et al.), and a different feldspar analysis (sample S1-13a, Supplementary Data). It is very annoying to have to dig in cited papers' supplementary materials to find this out, as the exact picked-up data-sets are not referenced by the authors. Sample S1-13a is an outlier amongst 16 undersaturated feldspar analyses in totals, most of them not displaying deviations from mass-dependent fractionation line. This is also true for olivine analysis O-01b, which despite having been chosen by Rzeplinski et al. 2022 for Fig.1, is not the nicest analysis they got judging from their Supplementary Data. What matters however is the statistical value of their entire data-set, as appreciated by Rzeplinski et al. reviewers, with most olivine and feldspar analyses showing clear mass-dependent fractionation.

Concerning the amplitude of the above-mentioned deviations from mass-dependent fractionation in these two selected samples, it is in fact much smaller than once reported in the authors own published data (Avice et al. GCA 2018, 124Xe on Fig.1b, and 126Xe on Fig.1c, see figure in the pdf file). The results of this study is central to the model of 'step by step' H+Xe+ escape from Earth's atmosphere, and as such is key in linking the Great Oxidation Event to the missing Xe paradox, as proposed by the authors. If we are to discredit a paper and its review process for a limited amount of individual data points a bit off of a regression line, then the model linking the Great Oxidation Event to the missing Xe paradox has to be discredited too? Or perhaps the authors have an insight on what causes these strong deviations which could also help interpreting the smaller deviations they point out in others' data-sets.

7) I 376-389: 'Most importantly, the progressive release of mantle Xe that had been repeatedly enriched in Xe relative to other noble gases (including Kr) is unable to produce an atmosphere with a Xe deficit, starting from a CI-like atmosphere.' Element fractionation works both ways, hence if Xe is preferentially incorporated in crystals upon magma ocean crystallization or during crystals/atmosphere equilibrium under pressure, the same is true for noble gas release: Xe is expected to be preferentially retained in crystals upon further partial melting or fluid release at depth. Only in case of complete degassing would be the statement by the authors true. The authors should discuss such cases or just remove the sentence.

There might also be a confusion here about the initial chondritic atmosphere that underwent mass-dependent fractionation in Rzeplinski et al. scenario, and the CI-like late veneer. It is the minor release of heavy trapped Xe in this CI-like late veneer atmosphere that is advocated in their scenario to explain atmospheric Archean evolution.

8) I 376-389: 'The proposed model would also require mantle Xe to be enriched in heavy Xe isotopes relative to the atmosphere, which is opposite to observations of light Xe isotope enrichments in mantle gases worldwide.' No, Rzeplinski et al. do not require mantle Xe to be enriched in heavy Xe isotopes relative to the atmosphere. As explicit in the Extended Data-Fig.4, mantle Xe in their scenario has the same Xe isotopic signature as the current-day atmosphere. The authors besides forget to mention here how Rzeplinski et al. explain the observation of light Xe isotope enrichments in mantle gases, i.e. by the contribution of recycled Archean atmosphere to the mantle source or by input from less or not fractionated lower mantle resulting from the last magma ocean stage having affected mostly/only the upper mantle and lower crust; these possibilities are also illustrated in the Extended Data-Fig.4.

9) 'At last, why the Xe isotope evolution of Earth's atmosphere (magmatic activity and continental erosion/metamorphism) would have stopped around the Great Oxygenation Event remains to be explained in the framework of this model.' This sentence implies that Xe isotope evolution abruptly stopped around the Great Oxygenation Event. However, judging from Fig.2a, it is not possible to resolve within the actual spread of data and associated error bars, if Xe isotope evolution decreased linearly throughout the Archean until to 2 Gy or if it was a parabolic decay, consistent with an equilibrium state reached around this time which also coincides with most of the continental crust being built up. Both possibilities should be discussed.

10) I.389-393: The mineralogy of the upper and lower mantle are different, and so far, Xe retention has only been reported in olivine (Crepisson et al. 2018), not in bridgmanite (Shcheka and Keppler 2012). The lower mantle is therefore not expected

to contain significant amounts of trapped Xe. Please be specific when mentioning deep mantle, i.e. upper mantle or lower mantle.

11) L.637: Nitrogen can also be retained as N₂ in hydrous silicates, as shown in cymrite in metapelites for instance (Sokol et al. 2020).

Reviewer #2 (Remarks to the Author):

My apologies, I was at the Goldschmidt meeting this past week.

The authors have done an excellent job revising this paper, I don't see further room for significant improvement and suggest it be published as it is. It will be highly cited and I intend to use it in my upcoming class on "origin and evolution of Earth's atmosphere."

Reviewer #3 (Remarks to the Author):

I am satisfied with the authors' responses to my overall minor comments. I have also read the other two reviewers' comments and the authors responses. I am impressed with the breadth and depth of the conversation. However, my impression is that the authors could have avoided much of their effort in adding content on "other volatile elements" by just revising the title to focus on noble gases and N₂. After all, only noble gases and N₂ can be faithfully preserved in fluid inclusions.

** Visit Nature Portfolio's author and referees' website at www.nature.com/authors for information about policies, services and author benefits**

Communications Earth & Environment is committed to improving transparency in authorship. As part of our efforts in this direction, we are now requesting that all authors identified as 'corresponding author' create and link their Open Researcher and Contributor Identifier (ORCID) with their account on the Manuscript Tracking System prior to acceptance. ORCID helps the scientific community achieve unambiguous attribution of all scholarly contributions. You can create and link your ORCID from the home page of the Manuscript Tracking System by clicking on 'Modify my Springer Nature account' and following the instructions in the link below. Please also inform all co-authors that they can add their ORCIDs to their accounts and that they must do so prior to acceptance.

Version 2:

Decision Letter:

Dear Dr BEKAERT,

Your revised manuscript titled "Fluid inclusions: tiny windows into global paleo-environments" has now been seen by our reviewers, whose comments appear below. In light of their advice we are delighted to say that we are happy, in principle, to publish a suitably revised version in Communications Earth & Environment.

We therefore invite you to revise your paper one last time to address the remaining concerns of our reviewer 1. At the same time we ask that you edit your manuscript to comply with our format requirements and to maximise the accessibility and therefore the impact of your work.

EDITORIAL REQUESTS:

*****Please take care to match our formatting and policy requirements. We will check revised manuscript and return manuscripts that do not comply. Such requests will lead to delays. *****

SUBMISSION INFORMATION:

OPEN ACCESS:

Communications Earth & Environment is a fully open access journal. Articles are made freely accessible on publication. For further information about article processing charges, open access funding, and advice and support from Nature Portfolio, please visit <https://www.nature.com/commsenv/open-access>

Link Redacted

Best regards,

Alireza Bahadori, PhD
Senior Editor
Communications Earth & Environment
Consulting Editor
Communications Sustainability

REVIEWERS' COMMENTS:

Reviewer #1 (Remarks to the Author):

The revised manuscript provides a fairer analysis of the literature on the missing Xe paradox, if not an entirely unbiased one. Readers should be able to form their own opinion, provided the final points below are taken into account.

1) I.290-292: 'However, it is now clear that none of these hypotheses is geochemically sound (BOX 2), as they fail to explain both the depletion of Xe relative to Kr in Earth's atmosphere, and, most critically, the isotopic evolution of atmospheric Xe throughout the Archean (Figure 2).'

This still needs to be softened up. There is no evidence here to suggest that all hypotheses concerning the storage of Xe at depth fail to explain the isotopic evolution of atmospheric Xe throughout the Archean. However, these hypotheses do face challenges, and this could be written as such.

2) I.377: 'several' should be changed to 'some', since no more than two potential caveats are discussed below (see the point below regarding the third one, which is not straightforward).

3) I.385-387: In their rebuttal letter, the authors agree that it is not possible to determine from the actual data and associated error bars whether Xe isotope evolution decreased linearly throughout the Archean until 2 Gy, or whether it decayed in a parabolic manner, which would be consistent with an equilibrium state being reached around this time, when most of the continental crust was formed. They have corrected the manuscript on lines 208–213, but not at the end of Box 2, where they repeat the argument. Either remove this argument or repeat I. 208–213.

4) I.747-749: While the authors acknowledge the advantages of their escape scenario, they also recognise that it faces

challenges, particularly the absence of a physical vertical transfer process. Therefore, it cannot be concluded that the missing Xe paradox has been solved. This sentence should be changed to: 'Coupling Xe isotope analyses in ancient fluid inclusions with modelling of Xe photochemistry in the atmosphere could help to solve the missing Xe paradox, provided that a physical process for lifting Xe throughout the atmosphere is identified.'

** Visit Nature Portfolio's author and referees' website at www.nature.com/authors for information about policies, services and author benefits**

Dear Dr BEKAERT,

We apologize for the delay in sending our decision letter.

Your manuscript titled "Fluid inclusions: tiny windows into global paleo-environments" has now been seen by 3 reviewers, whose comments are appended below. You will see that they find your work of some potential interest. However, they have raised quite substantial concerns that must be addressed. In light of these comments, we cannot accept the manuscript for publication but would be interested in considering a revised version that fully addresses these serious concerns.

We hope you will find the reviewers' comments useful as you decide how to proceed. Should additional work allow you to address these criticisms, we would be happy to look at a substantially revised manuscript. If you choose to take up this option, please either highlight all changes in the manuscript text file or provide a list of the changes to the manuscript with your responses to the reviewers.

In particular, please ensure that the revised manuscript meets the following editorial requests:

A restructuring of the manuscript is necessary to expand the conceptual framework of the research and broaden the discussions by citing relevant and recently published articles.

Specifically, the approach and considerations on noble gases should be compared to similar approaches, and the scientific debate on relevant topics should be acknowledged and discussed in depth. Additionally, the state-of-the-art analytical developments in the field should be expanded, and recent literature that informs the understanding of the ancient atmosphere using different geochemical tracers should be incorporated, as recommended by the reviewers.

Please do not hesitate to contact us if you have any questions or would like to discuss the required revisions further. Thank you for the opportunity to review your work.

Best regards,

Carolina Ortiz Guerrero, Ph.D.
Associate Editor
Communications Earth & Environment

Dear editor,

Apologizes for the delay in sending our revision.

We thank the editor for their careful handling of our manuscript. We understand that the original version of our manuscript was somehow biased towards discussing noble gas systematics to the expense of other volatile elements. We appreciate the constructive comments from the

reviewers, which led us to significantly revise our manuscript and broaden its scope. We have now included all the references suggested by the reviewers, which we supplemented with additional references on the relevant topics. Following the reviewers' comments we also significantly expanded the discussion of major volatile systematics and solubility effects. Altogether, the revisions led us to extensively restructure parts of our manuscript. As shown below in red, we have endeavored to adequately address every single comment from the reviewers.

We hope the manuscript will now be deemed suitable for publication in *Communications Earth & Environment*.

David Bekaert
On behalf of the co-authors.

REVIEWER COMMENTS:

Reviewer #1 (Remarks to the Author):

This article aims at reviewing what has been learnt from hydrothermal fluid inclusions on the evolution of paleo-environments, i.e. paleo-atmosphere and paleo-oceans. Indeed, much has been done, and this topic is of interest to a large community in Earth & planetary sciences.

The manuscript however mostly covers previous findings by the authors and is unbalanced in two ways: 1) nearly half of it is focused on the missing Xe paradox and the authors own scenario, 2) some references are missing or misleadingly cited, others are rejected without discussion or based on arguments that the reader cannot evaluate, being given only a very one sided view. This is quite far from a healthy scientific debate, as expected in a review paper.

We appreciate the reviewer taking the time to provide a detailed and constructive review of our work. We understand that the original version of our manuscript was somehow biased towards our own view, and so we have now strived to provide a more general literature review. We have tried to fix all of the instances where citation was not appropriate. Regarding the discussion of the Rzeplinski model, we originally decided not to dive into the intricacies of this paper. Following reviewer #1's suggestions, we have now added a figure in BOX 2 to properly discuss this model. We now expose the pros of the Rzeplinski model, before outlining its failures, based on scientific arguments. We hope that this will be fine with the reviewer.

The article also contains 4 boxes to give a pedagogic focus on some items, but 3 of them are oversimplified and miss key aspects. For these reasons, I cannot recommend publication of this article.

We have now endeavored to fix the boxes. However, for the sake of the review paper some aspects of the topics addressed here must be somewhat simplified. Some of the "details" required by the reviewer may unfortunately be beyond the scope of this manuscript, and so we had to make some choices to find a good balance between clarity, comprehensiveness, and exhaustivity. We hope that the substantial modifications of our paper will now lead the reviewer to recommend publication of our contribution.

More detailed comments:

1- A minimum of information on fluid inclusions in quartz (or other minerals) should be given, their geological context, type of host rocks, proper description of fluid inclusions distribution. For many readers, the title might be misleading as fluid inclusions are widely studied in magmatic/volcanic contexts. This article targets one specific type of fluid inclusions, which is fine of course but should be stated somewhere to avoid confusions.

We fully agree with the reviewer. We have now added several sentences to the introduction in order to provide the required information on geological context, type of host rocks, typical fluid

inclusion distributions “Fluid inclusions represent remnants of ancient geofluids that have been trapped in hydrothermal minerals (e.g., quartz, baryte). These hydrothermal minerals are typically found in so-called epithermal veins (low to intermediate temperature hydrothermal systems) formed by the intrusion of mineralizing fluids through volcanic, sedimentary, or intrusive units within shallow crustal levels. The distribution of fluid inclusions in hydrothermal minerals depends on several factors including crystal growth, deformation, and post-crystallization processes. In particular, populations of primary fluid inclusions (formed during crystal growth) are typically aligned along growth zones, trapped in clusters or isolated within the mineral lattice, whereas secondary inclusions preferentially form along healed fractures or microcracks after the mineral has crystallized.” L39-49.

2- Box 1: 'two populations of fluid inclusions, i.e. two different sizes, would result from two different events'. Such a description of fluid inclusions does not correspond to the reality and wrongly gives the impression that simple crushing steps can separate populations. Fluid inclusions in hydrothermal quartz grains are typically less than few tens of μm , and a single mineral grain contains several generations of fluid inclusions with distinctly different compositions.

The purpose of this box was to present the rationale behind using a step-crushing extraction protocol. If fluid inclusions of two generations have similar sizes and are distributed homogeneously within a sample, step-crushing will indeed not be able to separate these two generations of fluid inclusions. We now highlight that this is an oversimplification for the sake of clarity “Such a description of fluid inclusions is intentionally oversimplified, as fluid inclusions in hydrothermal quartz grains are typically less than a few tens of micrometers in size. Consequently, a single mineral grain may contain several generations of fluid inclusions with distinctly different compositions that may not be easily separable during the crushing process.” L111-115

3- p.116: 'over geological time, the composition of the atmosphere results from a balance between volcanic outgassing and ingassing to the solid Earth via subduction (Bekaert et al. 2021)'. This is mostly true for CO_2 , but not for other volatiles, noble gases included, for which volcanoes are just one way to outgas solid Earth, erosion and metamorphism are other essential processes (see Bender et al. 2008 for Ar for instance). This requires much more discussion as it is quite key to the formation of fluid inclusions in minerals, and the discussion should be supported by more references than just one by the author.

We agree. We have now changed the text to include the reviewer’s point “. Over geological time, the composition of the atmosphere results from a balance between outgassing of the solid Earth (mantle and crust) and ingassing into the solid Earth via subduction (Bekaert et al. 2021; Gibson & McKen 2023). While volcanoes are a major pathway for outgassing, especially for CO_2 , other volatiles – including noble gases – are also significantly released to the atmosphere during cryptic degassing, erosion and metamorphic processes (e.g., Bender et al. 2008).” L149-155.

4- 1.132: 'Noble gases are inert'. Further down the authors cite Grochala 2007, a review paper on noble gases compounds. It should be specified here that noble gases (xenon mostly) may not be inert at all conditions. The literature on the topic is quite extended, with several reviews more recent than Grochala 2007.

We understand the reviewer's point. We have now added a couple sentences to specify that under some circumstances the heavy noble gases can take part in forming chemical compounds. Reference to Grandinetti (2018) has also been added.

“Noble gases are considered inert under most terrestrial conditions, meaning they are not influenced by chemical or biological processes. Under ionized conditions or at high temperature/high pressure in the deep Earth, however, the enhanced reactivity of heavy noble gases (especially Xe) makes it possible for these elements to be incorporated into molecules and mineral phases (Grochala 2007; Grandinetti 2018). Under neutral ambient conditions at Earth's surface, the concentrations and isotope compositions of noble gases dissolved in paleo-fluids are controlled solely by physical and nuclear processes, which can be modeled and quantified (e.g., **Ballentine et al. 2002**).” L170-177.

5- 1.151. 'Unlike lighter krypton, xenon is easily ionized by solar UV or charge exchange with H⁺ ions, so Xe⁺ can be dragged out to space by escaping H⁺ ions without significantly affecting atmospheric Kr.' This statement is not that easy to follow since Xe ionisation energy (12.13 eV) is only slightly lower than that of Kr (14 eV). Why a small difference would imply such a Xe-specific depletion?

Indeed, the low ionization energy of Xe is not the only explanation for its efficient escape. This topic has been discussed in great detail by Zahnle et al. (2019): recombination of Kr ions into neutral Kr by interaction with hydrogen is fast while this process is almost nonexistent for Xe ions (Anicich, 1993). This means that even if some Kr is ionized in the early Earth's atmosphere, it is quickly converted back to neutral Kr and thus not available for escape in an ionized hydrogen wind.

6- 1.153 'Ionized Xe could then have then been being dragged along open magnetic field lines and lost to space via ionic coupling with escaping H⁺ (Zahnle et al., 2019) or becoming trapped in organic hazes formed within the CH₄-rich early atmosphere (Hébrard & Marty, 2014).' Concerning the first hypothesis, the authors must mention that Zahnle et al. do point out that so far, no mechanism to lift Xe (up to high altitudes at which it could ionise) has been found. This is the most problematic aspect of the Xe-escape scenario that must be clearly spelled out.

We agree with the reviewer. This is now specified L202-207 “ One major challenge of this atmospheric Xe escape model is that vertical atmospheric transfer is required to sustain the upward transport of Xe ions through the molecular ionosphere to the base of the outflowing

hydrogen corona and to maintain Xe escape (Zahnle et al. 2019). The nature and timescales of the mechanisms driving such vertical transfer remain to be described.”

Concerning the second hypothesis, the existence of such organic hazes is possible but is speculative.

Agreed. We have added “possibly” to this sentence to emphasize it is just a possibility “becoming trapped in organic hazes possibly formed within the CH₄-rich early atmosphere (Hébrard & Marty, 2014).”L201.

7- As pointed out by Dauphas, not cited here, whatever process lead to Xe fractionation and depletion affected Mars and Earth similarly, and hence cannot be related to mass/gravity as it differs a lot between Mars and Earth.

We agree but Mars and Earth atmospheric Xe are similar only at the first look. Zahnle et al. (2019) pointed out that the depletion is not exactly the same for two planets (Mars Xe 50% less depleted than Earth’s atmospheric Xe) and that the extent of isotopic fractionation is also significantly different (2.5 % per amu for Mars relative to Solar and about 4% per amu for Earth Xe relative to U-Xe). This suggests that the escape process acted differently on two planets. Gravity cannot be the only explanation here (Mars being smaller than Earth is expected to have lost more of its Xe) and this is one argument for advocating a non-thermal escape process in which ions are involved.

This is now specified L259-276: “Interestingly, atmospheric xenon on Mars also presents a mass-dependent enrichment in heavy Xe isotopes (Conrad et al., 2016). The extent of Xe isotope fractionation in the Martian atmosphere (2.5 %/amu relative to a solar initial composition) has long been considered smaller than that on Earth (~3.8 %/amu relative to U-Xe; Pepin 2000). However, more recent estimates of present-day terrestrial atmospheric Xe isotope fractionation, relative to its initial precursor, revise this value down to 2.6 %/amu, comparable to that of Mars (Cassata 2025). Measurements of trapped Martian atmospheric xenon in Martian meteorites also suggest that the isotopic fractionation of atmospheric xenon was still ongoing 4.4 Ga ago but that the final amplitude was established as soon as 4.1-4.2 Gyr ago (Cassata et al., 2022). Recently, Shorttle et al. (2024) proposed that energetic collisions which happened on early Mars (200-300 Ma after solar system formation) were powerful enough to drive atmospheric escape of xenon. It remains unknown if this process was also efficient on the Hadean Earth. Zahnle et al. (2019) also noted that the relative Xe depletion differs between the two planets, with Xe in the Martian atmosphere being ~50 % less depleted than in Earth’s atmosphere. This suggests that the Xe escape process operated differently on the two planets, and that gravity is not the dominant factor in atmospheric escape (as Mars, being smaller, would be expected to have lost more Xe). Instead, this points to a non-thermal escape process involving ion loss.”

8- 1.227: 'noble gases lighter than Xe did not experience significant escape to space (Ozima &

Podosek 1999; Pepin 2006).¹ Ozima and Podosek did not report that, they reported that only Xe was missing, not that only Xe was lost to space.

We agree. This sentence was rephrased as: “That the analysis of noble gases trapped in Archean rocks revealed no significant evolution in atmospheric Kr isotopes over time is consistent with the fact that only Xe eventually went missing (Ozima & Podosek 1999).” L290-292.

9- 1.231: No references are given for the thermal escape process (Jeans escape).

Reference to Hunten, D. M., Pepin, R. O., & Walker, J. C. 1987, *Icarus*, 69, 532 (10.1016/0019-1035(87)90022-4) has now been added.

10- Box 2. 1.263: Shcheka & Keppler 2012 do not report Xe potential for enhanced reactivity but a high retention of Ar in lower mantle bridgmanite (still named perovskite at the time), unlike for Xe.

Agreed, we have now changed the text: “Previous attempts to explain missing Xe have proposed either a diminished or enhanced reactivity of Xe with deep Earth minerals. For example, the significantly higher solubility of Ar compared to Xe in MgSiO₃ perovskite led Shcheka and Keppler (2012) to propose that Xe could have been depleted early in the Earth's lower mantle during the crystallization of perovskite from a magma ocean, relative to the lighter noble gases. A Xe-enriched primordial atmosphere could have subsequently escaped to space, leaving later mantle degassing to replenish Ar-Kr, and to a lesser extent Xe, in the atmosphere. As with all the models discussed below, this hypothesis fails to explain the isotopic evolution of atmospheric xenon observed throughout the Archean eon. » L327-335

1.269: Sanloup et al. 2005 investigated the retention of Xe in quartz. The scenario by Rzeplinski et al. answers most if not all questions raised in the first half of the manuscript, i.e. elemental and isotopic Xe depletion during the Hadean, and later evolution of the atmosphere by degassing of heavier Xe isotopes initially trapped in the crust/mantle rocks with a systematic preferential release of lighter noble gases (hence of Kr) throughout the Archean. They also provide a thorough analysis of Xe enrichment in the natural rock record (and not just in one instance as done here on 1.268) and to which extent it reflects the actual xenon content in rocks at depth due to retrodiffusion upon rock ascent as for any other volatile element. In other words, this scenario cannot be dismissed so lightly but should be properly discussed.

We have now significantly expanded our description of the Rzeplinski et al. scenario: “Recently, Rzeplinski et al. (2022) measured the isotopic composition of Xe trapped in synthetic feldspar and olivine formed during high-pressure crystallization experiments using air- or nitrogen-diluted Xe and Kr. These authors reported significant mass-dependent isotope fractionation (up to +2.3‰ per atomic mass unit), accompanied by strong Xe enrichment relative to Kr. These findings were interpreted as evidence that Earth's modern Xe isotope signature reflects repeated (between 9 and 15 events) interactions between a reduced primary

atmosphere and oxidizing, crystallizing magma oceans. In each episode of isotopic equilibration, heavy Xe isotopes would have been preferentially sequestered in deep silicate phases with elevated Xe/Kr ratios. Atmospheric loss driven by high-energy impacts and hydrodynamic escape would have selectively removed the successive atmospheres enriched in lighter Xe isotopes. A late influx of chondritic material toward the end of Earth's accretion would have then reset the atmosphere to a CI-like composition, which would have then evolved during the Archean through partial remobilization of heavy Xe previously trapped in the mantle and crust. This process would have produced the observed transition from a CI chondrite-like to a present-day atmosphere, as observed from fluid inclusion analyses (Avice et al. 2018).” L361-376

Likewise, we expanded the discussion of this scenario, and hope that this will now be fine with the reviewer: “However, this complex scenario suffers from several problems and caveats. While experimental results showed marked deviations relative to the starting isotopic composition, observed isotopic variations did not necessarily follow expectations from mass-dependent fractionation. Most importantly, the progressive release of mantle Xe that had been repeatedly enriched in Xe relative to other noble gases (including Kr) is unable to produce an atmosphere with a Xe deficit, starting from a CI-like atmosphere. The proposed model would also require mantle Xe to be enriched in heavy Xe isotopes relative to the atmosphere, which is opposite to observations of light Xe isotope enrichments in mantle gases worldwide. At last, why the Xe isotope evolution of Earth's atmosphere (magmatic activity and continental erosion/metamorphism) would have stopped around the Great Oxygenation Event remains to be explained in the framework of this model. The proposal that repeated interactions between the early Earth's atmosphere and its silicate reservoirs led to a missing and isotopically fractionated atmospheric xenon after several magma ocean episodes can therefore largely be questioned.” L376-389

11-1.280 'most of their analyses revealed isotopic variations that do not follow expectations from mass-dependent fractionation'. What kind of isotopic variations are the authors mentioning? This is a very unfair accusation that the readers cannot assess and considering that reviewers of Rzeplinski et al. 2022 most likely evaluated the reliability of the measurements.

We understand the reviewer's point. Nevertheless, despite the strengths of the peer-review process, it is not immune to shortcomings. An example of the isotopic variations that do not follow expectations from MDF, even when analytical errors are considered, is shown here below. However, we want this review to be as objective as possible, and so we have decided not to show or extensively discuss the Rzeplinski data in our paper. This figure below is just for the review process.

12- Box 3. The authors cite only one reference to support that outgassing of radiogenic ^{40}Ar into the atmosphere played a prominent role, while the role of the continental crust remained minor. This contrasts with the results from Bender et al. 2008 that point out the strong contribution from continents, Bender et al. is cited here but misleadingly (in particular in Fig.4 caption, box 3). As pointed out above, the review should be broader, discuss pros and cons of previous results, especially when they differ.

It is made very clear from our main text and Figure 4 that the composition of the atmosphere is modified by outgassing of both the mantle and continental crust. Nonetheless, we have now modified the caption of Figure 4 for “As a result, outgassing from the continental crust and mantle release high $^{40}\text{Ar}^*/^{36}\text{Ar}$ and $^{129}\text{Xe}^*/^{130}\text{Xe}$ gas that can modify the composition of the atmosphere through time (Bender et al. 2008; Marty et al. 2019). »

We also modified the main text to now explicitly mention that “For Ar, calculations by **Bender et al. (2008)** suggest that $\sim 75\%$ of the total $^{40}\text{Ar}^*$ outgassing from the solid Earth today ($\sim 1.12 \times 10^8$ mol/yr) originates from the continental crust ($\sim 0.85 \times 10^8$ mol/yr), with the remaining $\sim 25\%$ ($\sim 0.27 \times 10^8$ mol/yr) deriving from the mantle. For $^{129}\text{Xe}^*$, the formation of the continental crust arguably occurred after ^{129}I became extinct (i.e., after ~ 100 Myr after the formation of the Solar System), implying that virtually 100% of the outgassing $^{129}\text{Xe}^*$ originates from the mantle.” L460-465.

13- L.323: about ^{129}Xe deficit observed only in pre GOE atmosphere inclusions, it can indeed be explained by extensive outgassing in the late Archean (Marty et al. 2019), or it can be explained, as the overall Xe isotopic evolution, by the equilibrium reached between Earth's degassing (mantle and crust) and Xe ingassing at the end of the Archean.

Sorry, but we do not understand the reviewer's proposal. As shown in Figure 2 (below), available data suggest that a constant relative deficit of ^{129}Xe compared to the present-day

atmospheric composition existed prior to 2.5 Ga. The transition from this composition to the modern one appears to have occurred around the time of the Great Oxidation Event (GOE). The fact that the ^{129}Xe deficit remains constant before this time seems inconsistent with the reviewer's suggestion that the system was out of equilibrium prior to the GOE. Furthermore, xenon ingassing is thought to have become significant only after the GOE (e.g., Parai & Mukhopadhyay, 2018), so we do not understand how the ^{129}Xe deficit, observed only in pre-GOE atmospheric inclusions, could be explained by a proposed equilibrium between degassing and ingassing at that time. Therefore, no changes have been made to the manuscript.

14- L.394: 'attesting to the inefficient escape of nitrogen via fractionating escape mechanisms since that time'. How do the results of Marty et al. 2013 and Avice et al. 2018 compare with atmospheric escape models?

Results from Marty et al. 2013 and Avice et al. 2018 show that the isotopic composition of nitrogen did not change significantly ($\pm 5\%$) since the Archean. In the case of a fractionating escape mechanism such as the one operating on Mars, one would expect to have an increase of the quantity of heavy ^{15}N relative to light ^{14}N . This is why we state that there was no significant fractionation of atmospheric nitrogen by escape since the Archean.

15- The whole literature on volatiles recycling at depth through incorporation in minerals in subduction contexts is ignored (N_2 and Ar in particular).

One has to make choices when writing a literature review on such an extensive topic. We agree with the reviewer that a discussion about N_2 and Ar recycling efficiencies and their potential impacts on the use of $\text{N}_2/^{36}\text{Ar}$ data from fluid inclusions was missing. The text reported here below has been added to address, at least partially, this issue:

“While N_2 and Ar arguably have similar solubilities in basaltic melts (implying no significant fractionation of the $\text{N}_2/^{36}\text{Ar}$ during mantle degassing, at least at the current mantle oxidation state; Marty 1995), the atmospheric $\text{N}_2/^{36}\text{Ar}$ could have evolved through time due to the distinct recycling efficiencies of these elements during subduction. The recycling efficiencies of both elements during early subduction processes associated with Hadean and Paleoarchean plate tectonics (e.g., Shirey & Richardson 2011; Keller & Schoene 2018) would arguably have been low due to the high mantle temperatures (Figure 2b). The emergence of colder,

modern-style subduction (whereby a significant vertical volatile flux of surface materials to mantle depths) would have marked a pivotal shift in surface-mantle interactions (**Holder et al. 2019**), transitioning from a purely degassing state to a balance between global degassing and ingassing (i.e., recycling; **Bekaert et al. 2021**). While recycled Ar is primarily hosted in hydrated phases like serpentinites (and thus potentially lost to the mantle wedge during dehydration; **Kendrick et al. 2017**), N can be retained in metasediments and peridotites due to NH_4 substitution for K in phengite or N_3^- substitution for oxygen (**Watenphul et al. 2009**, **Cartigny & Marty 2013**), implying no significant loss of N by dehydration (**Busigny et al. 2011**; **Halama et al. 2014**). The recycling efficiencies of Ar and N into the mantle via subduction may thus differ significantly due to their distinct retention behaviors in subducted materials (**Bekaert et al. 2021**). While heavy noble gas (Ar, Kr, Xe) systematics clearly suggest that substantial recycling of surface volatiles into the mantle has occurred over the past ~2–3 billion years (**Holland & Ballentine 2006**; **Parai & Mukhopadhyay 2018**), the global recycling efficiency of nitrogen remains open to debate (**Labidi et al. 2020**; **Bekaert et al. 2021**), and the potential for past atmospheric $\text{N}_2/^{36}\text{Ar}$ variations remains uncertain.” L626-648

16- Section 4: it should be reminded here that the original signal might have been erased by later events; the high potential of misinterpreting Archaean hydrothermal systems from the investigation of fluid inclusions has been pointed out (Farber et al. 2015 for instance).

Agreed. This sentence has now been added to section 4: “The original geochemical signatures of fluid inclusions may be overprinted by later events (e.g., hydrothermal circulation associated with regional metamorphism), thus calling for caution when reconstructing paleoenvironmental conditions from fluid inclusion analyses (e.g., Farber et al., 2015).” L740-743. This information has also been added to Box 1.

Reviewer #2 (Remarks to the Author):

Paper does an excellent job on the noble gas part, but needs a bit more updated literature on the great data available using the bulk volatiles once they are re-interpreted more quantitatively.

Thank you for this constructive assessment of our work.

Review of Bekaert et al: Fluid inclusions: tiny windows into global paleo environments

This paper reviews the current state of our long-term understanding of the evolution of the atmosphere-ocean system from the perspective of the only direct samples available: mineral fluid inclusions. The paper does a nice, if brief, review of the noble gas literature, with a heavy focus on Xe. Although the history of Xe is among the most complicated of volatiles to understand, the authors show that its very complexity is what makes it potentially so useful despite the gaps in our current understanding. I think this review is compelling and well done. My main criticism is that I think some of the other noble gas data need to be explained in more detail, and I think the authors have missed an opportunity to highlight some very recent progress using information from the bulk volatiles in fluid inclusions to interpret the ancient atmosphere.

We agree that a discussion of other noble gas data and bulk volatiles in fluid inclusions was missing from our original submission.

1. My biggest criticism is that this paper is missing some review of the bulk atmospheric volatile evidence from fluid inclusions that informs our understanding of the major element composition of the ancient atmosphere. I understand that the current record of that is limited and what has been done in the past has been sloppy work (e.g, Brand et al 2021, Steadman et al 2020), but there have recently been some major advances into deconvoluting the atmospheric record from evaporitic bulk volatile ratios (c.f., Blamey et al 2016, vs. Park and Schaller 2024). This new work completely supports the argument the authors are trying to make and bolsters the “tiny windows” case and not including it is a missed opportunity. I would suggest at least a paragraph addressing these data, particularly because the older data (e.g., Blamey et al 2016, etc) gives the community the wrong impression of the capacity of fluid inclusions to give direct insights into the ancient atmosphere. Yes, the Park et al method involves using $^{40}\text{Ar}/\text{N}_2$ ratios as a partitioning factor, but those authors also mention the clear utility of the method using ^{36}Ar or another conservative tracer (e.g., another noble gas).

A new section about fluid inclusion phase chemistry and potential effects of solubility fractionation has now been added (see response to next question). We also added some information about previous work on evaporitic bulk volatile ratios: “Fluid inclusions trapped in evaporites (e.g., halite) also provide an opportunity to probe the gas composition of the ancient atmosphere, up to the Neoproterozoic period (e.g., **Blamey et al. 2016**). Over the past decades, great strides have been achieved in advancing techniques of gas extraction from halite fluid inclusions (e.g., heat or cold extraction followed by mass spectrometry analyses), making it

possible to increase resolution while reducing sample size requirements (Blamey, 2012). Critically, however, conservative tracers of gas-fluid partitioning (e.g., $^{40}\text{Ar}/\text{N}_2$; Park & Schaller (2025), Figure 5) are invariably required to provide robust insights into the ancient atmosphere's composition." L547-554. Thank you

2. Along the same lines, there needs to be a bit of discussion on the differences in solubility for the various gases and assumptions made about aqueous end members, etc. The authors are not concerned with solubility effects because they are looking primarily at noble gases, but there is even an isotope effect on e.g., Ar that should be accounted for (for example, see Seltzer, 2019). More importantly, N_2 and Ar have very different aqueous solubilities, but the authors interpretation of the N_2 vs. Ar contains a tacit assumption that they are looking at the fully aqueous endmember. E.g., the discussion on lines 408-416 would benefit from some mention of the effect of differences in solubility and how it could be reconciled.

We have carefully taken into account this comment, and added a whole section about bulk atmospheric volatiles. We hope that this will satisfy the reviewer, L493-528:

"2.1. Fluid inclusion phase chemistry.

While it has often been assumed that gas compositions measured from fluid inclusions can be directly interpreted as reflecting the atmospheric conditions at the time of mineral precipitation, there is growing evidence that solubility effects associated with the partitioning of volatiles between gas and aqueous phases present at the time of inclusion formation must be considered to avoid misinterpretation (e.g., Park & Schaller 2025). Because each gas has a different solubility in water, the composition of gaseous inclusions (e.g., trapped air bubbles) can differ markedly from that of the dissolved gases in entrapped fluids. For instance, while the modern atmosphere is composed of about 78.1% N_2 , 20.9% O_2 , 0.9% Ar, and ~ 420 ppm CO_2 , these proportions shift significantly upon dissolution in freshwater at 20°C, yielding a composition of about 63.0% N_2 , 33.4% O_2 , 1.6% Ar, and 1.9% CO_2 (Figure 5). These proportions depend on the salinity of the fluid phase and temperature of the system, yielding about 70.3% N_2 , 25.7% O_2 , 1.8% Ar, and 2.3 % CO_2 for seawater-like fluids at 90°C (Figure 5). As a result, elemental ratios such as N_2/Ar may vary from ~38 in fresh air-equilibrated water at 20°C, up to ~84 in pure air bubbles. Park & Schaller (2025) emphasized the importance of these considerations and proposed a robust approach using the N_2/Ar as a proxy for calculating the gas volume fractions (ϕ_g ; Figure 5) at the time of entrapment, therefore allowing the observed gas ratios to be corrected to accurately reflect the composition of the atmosphere under which the fluid inclusions formed. Recent studies have also shown that estimates of the paleo-atmospheric elemental ratio can be obtained from crushing experiments on ancient

hydrothermal quartz and baryte (Broadley et al. 2022; Avice et al. 2025), although the true Kr/Xe of the Archean atmosphere remains elusive given the number of processes able to impart elemental fractionation (Avice et al., 2025).

Figure 5: The gas composition of fluid inclusions is modeled as a mixture between modern atmospheric air and air-saturated water (modified from Park & Schaller (2025)). The resulting composition depends on the relative contribution of each endmember, governed by the gas volume fraction (ϕ_g). When $\phi_g = 1$, the inclusion reflects the atmospheric composition, when $\phi_g = 0$, it reflects the composition of dissolved gases in water. The wide range of possible CO_2 mole fractions is shown altogether with Ar on a logarithmic y-axis. Ranges of selected elemental ratios are shown on the right-hand side. All calculations using solid lines assume 20 °C freshwater, whereas dotted lines show calculations for 90°C seawater, for comparison.”

For example, the data shown in Fig. 6 has always intrigued me – how much of the X axis ($N_2/36Ar$) is due to the difference in solubility between N_2 and Ar? I think the community is finally starting to realize that interpreting these ratios on a gas basis or aqueous basis is not totally accurate. Marty et al assume that the inclusions are all aqueous and that the Air end-member is actually ASW. This is probably a fair assumption, but if it were dominated by gas the assumption could impart a factor of 2.5 in the x-axis of this plot if there is a gaseous component unaccounted for. Unless this can be demonstrated in another way independently, I think we need to interpret those data with more care. This is especially true because the $N_2/36Ar$ of 1 atm air-saturated-brine overlaps with the “2-3 times p N_2 ” bounds: the $N_2/36Ar$ of Brine

is $1-1.3 \times 10^{-4}$ depending on the temperature and salinity, but the $N_2/^{36}Ar$ of modern Air would be 2.5×10^{-4} , which completely overlaps with the “2-3 times pN₂” bar in Fig. 6.

We understand the criticism. This is now discussed L 607-624 “However, existing measurements often face difficulties in clearly determining the composition of the paleo-atmospheric endmember and the value of paleo-atmospheric pN₂ (Figure 7). This is due to the ubiquitous presence of a hydrothermal component rich in crustal N₂ (high N₂/³⁶Ar) and radiogenic ⁴⁰Ar* (high ⁴⁰Ar/³⁶Ar; Nishizawa et al., 2007). For the Marty et al. (2013) dataset, for example, the analyzed fluid inclusions arguably represent hydrothermal fluids derived from seawater (Foriel et al 2004; Thébault et al. 2006). While the occurrence of air bubbles can be discarded based on the intra-ocean origin of the host minerals, a contribution from magmatic CO₂ cannot be eliminated. Based on mass balance calculation, however, the N₂/Ar ratio measured from fluid inclusion would still represent that of air-equilibrated water rather than a magmatic component. Future measurements on samples with a higher relative proportion of the paleo-atmospheric component (Avice et al. 2023) might help determine the pN₂ of the ancient atmosphere with greater precision. That said, there exist lingering uncertainties about whether a direct link can be established between the measured N₂/³⁶Ar ratio of the gas released from fluid inclusions and the “true” atmospheric N₂/³⁶Ar ratio (Marty et al. 2013; Avice et al. 2018). Analysis of clumped N₂ (Yeung et al. 2017) from ancient fluid inclusions represents an analytical challenge that would offer a promising avenue for distinguishing atmospheric-derived and hydrothermal nitrogen.”

3. Step crushing has been a useful method in the noble gas community for many years. I particularly appreciate the box insets describing the ability of this technique to progressively reveal the history of volatiles stored in different generations of inclusions. However, as a practitioner of this method myself, I am often confronted with the alternative ‘step-heating’ method as in some way superior. I think the authors should highlight that (outside of the noble gas realm) step heating is undesirable for the study of ancient volatiles because it can create or induce chemistry that would otherwise not have occurred within the samples (maybe they say this but I missed it).

This is now specified L92-95: “This approach is preferred to any ‘step-heating’ method for the study of ancient volatiles in fluid inclusions, as heating may generate undesirable chemical reactions (except for noble gases) that would otherwise not occur within the samples.”

Yes, N₂ is treated as essentially inert, but ²⁸N vs ²⁹N have different solubilities (effect is ~1%), which are a function of temperature.

This is now specified L 576-577: “Note that potential isotopic effects related to phase chemistry (i.e., difference in solubility between ¹⁵N and ¹⁴N) are minor, lower than the part per thousand (permil) level (Park & Schaller 2025).” L591-592

Also, step heating classically much better at accessing the molecular domains where e.g., ^{40}Ar has accumulated from decay within the crystal lattice unconnected to inclusions or other defects. It is a transition from solid to gas that is then liberated by diffusion/total fusion. Step crushing can only access small amounts of this domain (hence the small amount of $^{40}\text{Ar}^*$) in the crush data, but the crush data are primarily dominated by Ar in solution or from $\text{K} \rightarrow \text{Ar}$ decay in solution/gas inclusions.

Right. This is now specified L133-138: “Accessing volatiles from mineral domains that are unconnected to inclusions or other defects (e.g., radiogenic ^{40}Ar produced by in-situ decay of ^{40}K within the mineral lattice) requires heating to release the volatiles by diffusion, mineral breakdown, or total fusion. The classical approach, therefore, is to first extract volatiles from fluid inclusions via step crushing, followed by step heating and eventual fusion of the crushing residue (e.g., Almayrac et al. 2021).”

60-62 – this is not totally true, though I understand the authors perspectives here. There is a difference in solubility between ^{40}Ar and ^{36}Ar . I suspect it does not matter much for the points the authors are trying to make but it effects the 40/36 ratio at the tenths place (0.6 to 0.9).

Right, ϵ_{sol} is on the order of 1‰ for $^{40}\text{Ar}/^{36}\text{Ar}$, which is small compared to the analytical precision of static mass spectrometers. This is now specified as “Importantly, however, isotope fractionation resulting from differences in the solubility of various isotopes is minimal compared to the typical precision of mass spectrometers (Seltzer et al., 2023).” L71-73.

197-200 – its also possible there was no appreciable field present (e.g., Tarduno 2021)

This comment refers to the sentence : « One possibility is that Xe escape was confined to polar regions due to opening of the geomagnetic field, driven by sporadic bursts of intense solar activity, restricted to short periods with abundance hydrogen, or a combination of all these factors »

We have added reference to Tarduno but prefer not to discuss this complex topic in great detail: “Although the characteristics of Earth’s magnetic field during the Hadean and Archean eons remain uncertain (Tarduno et al. 2025), one possibility is that Xe escape was confined to polar regions due to opening of the geomagnetic field, driven by sporadic bursts of intense solar activity, restricted to short periods with abundance hydrogen, or a combination of all these factors.” L250-254

243-245 – e.g., why do we not see this with Ar?

The corresponding sentence is : “Unlike Xe, Kr^+ ions are neutralized by reaction with H_2 , therefore explaining the lack of Kr isotope fractionation in the ancient atmosphere” L304-305.

Kr is indeed more difficult to ionize than Xe. In addition, Kr^+ interacts with hydrogen and charge exchange neutralizes Kr^+ into Kr. The reaction with O is also very efficient (Zahnle et

al., 2019). Argon is more difficult to ionize than hydrogen. The consistent $^{38}\text{Ar}/^{36}\text{Ar}$ ratios from paleo atmospheric data are also consistent with no significant escape.

509-510 – but there are ways to put direct constraints on $p\text{CO}_2$ from fluid inclusions, why not mention them here?

We are not sure of which study the reviewer is referring to. Ancient $p\text{CO}_2$ have been determined by studies of paleosols, but these data are heavily debated, as discussed by Catling & Kasting, 2017, 10.1017/9781139020558.002). Catling and Zahnle (2020) also proposed a summary of available constraints:

>0.0004 bar (0°C), >0.0025 bar (25°C), or >0.26 bar (100°C) from 3.2 Ga Siderite weathering rinds on river gravel (Hessler et al. 2004)

0.03–0.15 bar from 2.77 Ga Mt. Roe paleosol, Australia (Kanzaki & Murakami 2015)

0.02–0.75 bar from 2.75 Ga Bird paleosol, South Africa (Kanzaki & Murakami 2015)

0.003–0.015 bar from 2.69 Ga Alpine Lake paleosol, MN, United States (Driese et al. 2011)

0.05–0.15 bar from 2.46 Ga Pronto/NAN paleosol, Canada (Kanzaki & Murakami 2015)

<~0.8 bar at 3.8–2.4 Ga in order to have enough UV to make S-MIF (Farquhar et al. 2001)

However, most of these constraints do not come from fluid inclusion analyses and so we prefer not to discuss them.

Fig. 6: this data has a very complex interpretation which needs to be made clear in the text, it is not at all self-evident how the partial pressure of N_2 is determined using this approach/data. Also, much of the data from Marty et al 2013 is from step heating

We understand this criticism, and this is why Avice et al. (2018) GCA proposed an upper bound for the partial pressure of N_2 in the atmosphere, rather than an absolute estimate. Taken together, all of our data suggest that the $p\text{N}_2$ was not significantly greater than modern during the Archean eon, without providing stringent constraints on how low the $p\text{N}_2$ could have been. Note that most data from Marty et al. 2013 are from step crushing and not heating.

In order to make this clear, we have now added this text : ” In detail, **Marty et al. (2013)** proposed that the $p\text{N}_2$ during the Archean was similar to the modern value. In a subsequent study, using new samples that exhibited well-defined correlations in the $^{40}\text{Ar}/^{36}\text{Ar}$ vs. $\text{N}_2/^{36}\text{Ar}$ space, **Avice et al. (2018)** confirmed that Archean $p\text{N}_2$ was arguably not higher than the modern

value and further suggested it may have been lower. These data do not put any stringent constraint on how low the pN_2 could have been during the Archean eon. There exist lingering uncertainties about whether a direct link can be established between the measured $N_2/^{36}Ar$ ratio of the gas released from fluid inclusions and the "true" atmospheric $N_2/^{36}Ar$ ratio (Marty et al. 2013; Avice et al. 2018), and additional work is therefore needed to better constrain the exact pN_2 of the Archean atmosphere." L600-609.

Fig. 3, Fig. 5, and Fig. 6: I suggest keeping the color scheme and symbology the same for the same authors and data sets to be consistent.

Thank you for this comment. We have revised the figures to make sure that the color scheme and symbology are consistent.

Reviewer #3 (Remarks to the Author): [See attached]

This manuscript provides a review on the evolution of Earth's lithosphere and atmosphere based on information obtained from fluid inclusions in minerals. The focus is on noble gas data. The review is timely and accurate.

I have only two minor suggestions:

1. I would suggest adding the state-of-art analytical development in the field, including sampling strategies, isotopic measurements, and analytical bottlenecks or areas for improvement

Thank you for this suggestion. We now added “Novel crushing techniques for fluid inclusion extraction (Vogel et al., 2013; Wilske et al., 2023), combined with ongoing analytical developments in nitrogen and noble gas mass spectrometry (Broadley & Bekaert, 2024) — including the emerging possibility of measuring all noble gases, and potentially nitrogen, in the same gas fractions (Nishizawa et al. 2007; Marty et al. 2013; Avice et al., 2023; Cattani et al., 2024) — hold great promise for advancing the use of fluid inclusions as tiny windows into Earth's past environments.” to the take-home bullet points.

2. Be specific when outlining a research paper, an idea, or an argument so that nonspecialist reader could understand.

We have reviewed our manuscript and done our best to make our purpose as clear and accessible as possible to non-specialists. Thank you.

Specifics:

Line 38: I would think “geofluids” is sufficient and there is no need to add “paleo-”.

Agreed, we have now removed “paleo”. The text now reads “Geofluids (waters, hydrocarbons, supercritical fluids, volatile gases) play a crucial role in many geological processes, ranging from molecular-scale fluid-rock reactions to global tectonics, and biological activity within Earth's crust (e.g., Fyfe 2012). Fluid inclusions represent remnants of ancient geofluids that have been trapped in hydrothermal minerals (e.g., quartz, baryte).”L37-41

Line 151: I suggest after “Xe has a low ionization potential relative to the other noble gases”. Perhaps add something like “making it easier to remove an electron from its outermost shell.”

Suggested change made L197-198

Line 204-206: The sentence or specifically “in favor of” is ambiguous. Does it mean that heavier Xe is leaving Mars preferentially or is it the opposite?

This has now been reformulated as “Interestingly, atmospheric xenon on Mars also presents a mass-dependent enrichment in heavy Xe isotopes (**Conrad et al., 2016**).” L259-260. Thank you.

Line 286-287: Change to “It leaves little doubt that they are incompatible.”

Suggested change made.

Line 290-293: The logic is hard to follow.

This was the original sentence: “Recent analyses of heavy noble gases in oceanic island basalts suggested that the deep mantle also displays a singular xenon depletion, which could have been acquired early in Earth's history, most likely during accretion (**Péron and Mukhopadhyay 2022**). As such, the solid Earth represents an unlikely reservoir to explain missing atmospheric Xe. » The idea is that, if missing atmospheric Xe were stored in the deep mantle, then a Xe excess may be expected for the analyses of rocks deriving from this reservoir. Rather, these authors suggest Xe is also depleted in the deep mantle, implying that the missing atmospheric Xe is not stored in this reservoir.

We have now reformulated this sentence to improve clarity: “Storing missing atmospheric Xe in a mantle reservoir would arguably require this reservoir to exhibit a Xe enrichment relative to other noble gases. Recent analyses of heavy noble gases in oceanic island basalts suggested that, rather than displaying a Xe excess, the deep mantle displays a singular xenon depletion probably acquired early in Earth's history (**Péron and Mukhopadhyay 2022**). As such, this reservoir does not contain missing atmospheric Xe. [...] One potential way to advance this problem would be to analyze noble gases in ancient (Archean or older) mantle rocks and minerals (e.g., diamonds), which represents a formidable challenge given the paucity of such samples, potential issues associated with their preservation over eons, and the analytical challenges associated with the analysis of such samples with low gas abundances.” L393-406

Line 345: miss a figure number.

Corrected.

Line 410-411: Difficult to understand.

This is the sentence: “This is due to the ubiquitous presence of a hydrothermal component rich in parentless radiogenic ^{40}Ar and in N_2 (**Nishizawa et al., 2007; Figure 1**).” This is now reformulated as “This is due to the ubiquitous presence of a hydrothermal component rich in crustal N_2 (high $\text{N}_2/^{36}\text{Ar}$) and radiogenic $^{40}\text{Ar}^*$ (high $^{40}\text{Ar}/^{36}\text{Ar}$; **Nishizawa et al., 2007**).” L609-610

Line 448-449: reorganize the sentence to eliminate ambiguity.

Sentence now reads “This also raises the possibility that significant amounts of oxygen were present in the Archean oceans, as oxygen solubility in seawater increases with decreasing salinity and temperature (**Weiss, 1970**).” L682-684 Thank you

Line 471-472: “relate to mantle and biological (i.e., organic matter) influences”. This is too vague in delivering useful information. It should be more specifically explained.

Agreed, the text has now been changed for “Because iodine exhibits a strong affinity for organic matter, higher iodine concentrations in the Archean Ocean compared to today could potentially reflect reduced biological sequestration, assuming the total organic reservoir was smaller than at present. This scenario may be consistent with the elevated Br/Cl ratios observed in Archean seawater (**Gutzmer et al., 2003**), although the variable influence of mantle-derived hydrothermal vent inputs cannot be ruled out (**Burgess et al., 2020**).” L708-713.

Line 29 & Line 510: It is not appropriate to have “Faint Young Sun Paradox” only mentioned in the abstract and in the conclusion.

Agreed. This is now discussed in the new section 2, “Major element composition of the ancient atmosphere”:

“Documenting the volatile element composition of the ancient atmosphere is key to understanding the environmental, geological, and climate events that accompanied early life evolution. One of the strongest constraints on Archean atmospheric composition is that the ground-level mixing ratio of O₂ was <10⁻⁶ PAL (present atmospheric level; **Figure 2**), and that the release of O₂ by early cyanobacteria as a byproduct of oxygenic photosynthesis dramatically changed Earth's atmosphere, transforming Earth's weakly reducing, anoxic atmosphere into an oxygenated one during the GOE, ~2.5 Gyr ago. Estimates of other Archean atmospheric gas concentrations are subject to significant uncertainty, with for example CO₂ and CH₄ levels ranging ~10 to 2500 and 10² to 10⁴ times modern amounts, respectively (**Catling & Zahnle 2020**). Interestingly, there exists an apparent contradiction between astrophysical models, which suggest that the Sun's luminosity was about 25–30% weaker during the Archean eon and so the Archean Earth should have been frozen, and geological evidence indicating that Earth's surface temperatures were warm enough to support liquid water and early life (**Feulner 2012**). This so-called faint young sun paradox is thought to be resolved by higher concentrations of greenhouse gases (such as CO₂, CH₄, H₂) in the early atmosphere (**Goldblatt & Zahnle 2011**).” L530-545

Dear Dr BEKAERT,

Your revised manuscript titled "Fluid inclusions: tiny windows into global paleo-environments" has now been seen by 3 reviewers, whose comments are appended below. You will see that the reviewers appreciate the effort you put in the revisions, but reviewer 1 continues to raise important concerns about factual mistakes and misconceptions. Given that these issues have been raised before and not fully addressed, we are not certain whether you are able to fully address the referee's concerns; if not, unfortunately we cannot consider your manuscript further, and would therefore recommend you seek publication elsewhere. In light of these ongoing concerns, although we cannot accept the manuscript for publication, we would be interested in considering a revised version that fully addresses these serious concerns. Specifically, for publication in *Communications Earth & Environment* to be appropriate, a revised manuscript must include a comprehensive comparative analysis of the literature on the missing Xe paradox, compellingly address atmospheric escape models, and fully incorporate the impact of deep Earth mineralogy on noble gas retention.

We hope you will find the reviewers' comments useful as you decide how to proceed. If additional work allows you to either incorporate or refute these criticisms, we will be happy to look at a substantially revised manuscript. If you choose to take up this option, please either highlight all changes in the manuscript text file, or provide a list of the changes to the manuscript with your responses to the reviewers.

When resubmitting, please provide a point-by-point response to the reviewers' comments. Please submit your responses as a separate file, distinct from your cover letter where you can add responses to the Editors' comments that you do not want to be made available to the reviewers. Word files are preferred. We recommend that any figures, tables or graphs that are included in the response to reviewers are also included in the main article or Supplementary Information.

If the revision process takes significantly longer than three months, we will be happy to reconsider your paper at a later date, as long as nothing similar has been accepted for publication at *Communications Earth & Environment* or published elsewhere in the meantime.

Please do not hesitate to contact us if you have any questions or would like to discuss the required revisions further. Thank you for the opportunity to review your work.

Best regards,

Alireza Bahadori, PhD
Associate Editor
Communications Earth & Environment
Consulting Editor
Communications Sustainability

Dear Editorial Board,
Dear Dr. Alireza Bahadori

Thank you for your careful handling of our manuscript and for the opportunity to respond to the reviewers' comments.

We note that two of the three reviewers are satisfied with both our responses to the reviews as well as with the current revised version of the manuscript, and suggest publication of our manuscript in its current state. **Reviewer 2** states there is “no further room for significant improvement” and considers the manuscript likely to be “highly cited.” **Reviewer 3** is “impressed with the breadth and depth of the conversation” and “satisfied with the authors' responses”.

Reviewer 1, by contrast, remains unsatisfied by our handling and criticism of the study by Rzeplinski et al. While we acknowledge their concerns (and are able to provide further edits to our manuscript), we highlight that discussion of this specific study represents only a small section of our broad review paper. Nonetheless, based on your editorial summary, we understand that a favorable recommendation would require us to follow **Reviewer 1**'s viewpoint and avoid criticisms of the Rzeplinski model. As demonstrated below, we have done our best to comply with your editorial recommendations. Although **Reviewer 1** views the critique of the Rzeplinski model as “central” to our manuscript, we emphasize again that it constitutes only a limited part of a much broader discussion.

We sincerely hope that the paper will now be considered ready for publication in its present form.

We thank you again for your editorial handling and guidance.

On behalf of the co-authors,
David V. Bekaert

Reviewer #1 (Remarks to the Author):

The revised manuscript has been corrected for many of the miscitations that were present in the first version of the manuscript, and is now less focussed on Xe, including a broader discussion of atmospheric gases.

Thank you. We agree with the reviewer that the paper is now significantly improved and has a much broader scope, with less emphasis on xenon. For this very reason, we find it unfortunate that the reviewer chose not to recommend publication based on our discussion of a single study, namely the Rzeplinski (2022) paper.

However, concerning Rzeplinski et al. 2022 whose criticism is quite central in the manuscript, while their scenario is more explicitly laid down now, there are still factual mistakes, misconceptions (see detailed comments below), and most concerning, the disturbing selection of only two analyses out of a whole data-set to build up their criticism. Some key references on the missing Xe paradox have not been added. Overall, the paper lacks a comparative analysis of the literature on the missing Xe paradox and its key characteristics: is there any experimental confirmation attesting Xe isotopic fractionation for each scenario (e.g. atmospheric escape models, accretion models, trapping at depth models)? Same question for Xe vs Kr elemental fractionation? Therefore, despite the improvements, I still cannot recommend it for publication at this stage.

Potential mistakes and misconceptions have been corrected (see below), and the argument concerning data quality – previously illustrated in our response document (not in the main text) – has now been removed in order to not have a discussion between authors and reviewers hindering the publication process. We have addressed every single comment from the reviewer, and we believe it is now important to recognize that the scope of our review extends well beyond the Rzeplinski paper. Following the reviewer's suggestions, we have given extensive attention to the discussion of the Rzeplinski paper, more than we originally thought would be reasonable. Sentences such as “The proposal that repeated interactions between the early Earth’s atmosphere and its silicate reservoirs led to a missing and isotopically fractionated atmospheric xenon after several magma ocean episodes can therefore largely be questioned. The cause(s) of these isotopic variations remain unknown, but it leaves little doubt that they are incompatible with the mass-dependent trend shown by atmospheric xenon relative to plausible starting compositions” have simply been removed from our manuscript, even though they truly reflect our thinking. We feel it is now unreasonable to (i) expect us to echo the reviewer’s own views about this paper – if they wish to do so, they are encouraged to write their own review–, and (ii) to prevent the publication of a comprehensive review article because of a disagreement over the choice of a particular dataset to illustrate a scientific point in a response document.

The references mentioned by the reviewer have been added when relevant to address their concerns, although some of them are outdated by more recent contributions. We have also included experimental evidence for Xe isotope fractionation during adsorption onto organics under ionized conditions, and we now clearly state that “no experimental evidence firmly supports the Xe escape hypothesis, as such a mechanism has not yet been reproduced in the laboratory”.

. Sentences like “The proposal that repeated interactions between the early Earth’s atmosphere and its silicate reservoirs led to a missing and isotopically fractionated atmospheric xenon after several magma ocean episodes can therefore largely be questioned. The cause(s) of these isotopic variations remain unknown, but it leaves little doubt that they are incompatible with the mass-dependent trend shown by

atmospheric xenon relative to plausible starting compositions” have simply been removed from our manuscript, even though they truly reflect our thinking.

We now hope that the reviewer will now deem our manuscript suitable for publication.

1) L.194: ‘Because Xe has a low ionization potential relative to the other noble gases’: As mentioned in the previous round of review, this needs a bit of context or quantification, please specify Xe and Kr ionisation potentials. This can alternatively be added later, on l. 311 along with the ionization potential of Ar.

Krypton and xenon ionization potentials have now been added L194: ‘Because Xe has a low ionization threshold (12.13 eV) relative to the other noble gases (e.g., 15.76 and 14.00 eV for Ar and Kr, respectively; making it easier to remove an electron from its outermost shell), atmospheric Xe could have been readily ionized by enhanced ultraviolet radiation”.

2) The comparison between Mars and Earth is now more deeply described as far as Xe isotopes are concerned. For Xe elemental depletion however, the authors mention Zahnle et al. 2019 who pointed out that the extent of Martian atmosphere Xe depletion is 50% that of the Earth’s atmosphere. However, this 50% variation is to be put into context, with atmospheric Xe being elementally depleted by a factor of 24 relative to Kr in CI chondrite. I still do not understand why none of Dauphas’s papers on the topic is cited, or his more broader 2014 article (Dauphas and Morbidelli, 2014).

The factor 24 (relative to Kr in CI) is outdated, and we prefer the range “10 to 20 times” already given L275 (e.g., Bekaert et al. 2020; Broadley et al. 2021). Note that the Dauphas & Morbidelli chapter in Treatise of Geochemistry (2014) is now somewhat outdated by Marty and Genda TOG (2024). However, reference to Dauphas & Morbidelli (2014) was added to satisfy the reviewer, L264-268: "It was historically proposed that Earth could have formed from Mars-like embryos, and that certain characteristics of the present-day atmosphere (e.g., atmospheric Xe isotope fractionation) could reflect processes that occurred on these early building blocks (see the review by **Dauphas & Morbidelli 2014**)."

3) L.323: This is supposed to be a review paper. Atmospheric escape models and models considering the effect of deep Earth mineralogy on noble gases retention should be discussed equally as mentioned in the first paragraph above, not siding blindly on the authors preferred scenario. Box 2 title should be objective, for instance ‘Missing atmospheric Xe: alternative scenarios’ with or without a question mark.

We have now changed the title of the box (“**BOX 2/ Missing atmospheric Xe: alternative scenarios?**”) following the reviewer’s recommendation.

4) L.357: Rzeplinski et al. 2022 used natural feldspars and olivines, not synthetic ones. Please correct.

This now reads “Recently, Rzepliński et al. (2022) measured the isotopic composition of Xe trapped in natural feldspar and olivine samples confined at high pressures and high temperatures with air- or nitrogen-diluted Xe and Kr.” L362-364.

5) L. 376-389: ‘*These findings were interpreted as evidence that Earth’s modern Xe signature reflects repeated (between 9 and 15 events) interactions between a reduced primary atmosphere and oxidizing, crystallizing magma oceans.*’

The mention of a reduced primary atmosphere and oxidizing magma oceans does not correspond to what is described in Rzeplinski et al. 2022. There is no mention of a net gain nor loss of oxygen in their scenario, just Xe substitution to Si in crystals at depth resulting in Xe oxidizing, not the whole magma ocean. The words ‘reduced’ and ‘oxidizing’ should be removed.

The caption of figure 3 in the Rzeplinski paper precisely mentions “multiple events of equilibrium between **reduced** Xe in a primary atmosphere and **oxidised** Xe in the crystallising magma ocean”. To avoid confusion, we have removed the words “reduced” and “oxidizing” following the reviewer’s suggestion.

6) L. 376-389: *‘However, this complex scenario suffers from several problems and caveats. While experimental results showed marked deviations relative to the starting isotopic composition, observed isotopic variations did not necessarily follow expectations from mass-dependent fractionation.’*

There is some improvement here compared to the first version of the manuscript, with a move from ‘that most of their analyses revealed isotopic variations that do not follow expectations from mass-dependent fractionation’ to ‘did not necessarily follow expectations from mass-dependent fractionation.’

We agree with the reviewer that we already made great effort trying to accommodate the reviewer’s view in the previous round of revision.

Nonetheless, as the authors claim to be objective in their answer to raised comments, the word ‘necessarily’ should be quantified. Do the majority of the analyses lie off a mass-dependent fractionation line or only selected outliers? The figure shown for the sake of the review process only, misleadingly leads the reviewers to think it is the original figure from Rzeplinski et al., which it is not (see figure in the pdf file). After careful inspection, it appears that the authors have chosen to reproduce olivine analysis O-01b (as for the original Fig.1, Rzeplinski et al.), and a different feldspar analysis (sample S1-13a, Supplementary Data).

Fig. from Bekaert et al., response to referee 1.

Fig.1 from Rzeplinski et al. 2022.

It was never our intention to pretend that this figure was the original figure from Rzeplinski et al.

In fact, it was clear from our review that we presented “an example of the isotopic variations that do not follow expectations from MDF ».

It is very annoying to have to dig in cited papers’ supplementary materials to find this out, as the exact picked-up datasets are not referenced by the authors. Sample S1-13a is an outlier amongst 16 undersaturated feldspar analyses in totals, most of them not displaying deviations from mass-dependent fractionation line. This is also true for olivine analysis O-01b, which despite having been chosen by Rzeplinski et al. 2022 for Fig.1, is not the nicest analysis they got judging from their Supplementary Data. What matters however is the statistical value of their entire dataset, as appreciated by Rzeplinski et al. reviewers, with most olivine and feldspar analyses showing clear mass-dependent fractionation.

Although the reviewer acknowledged that "There is some improvement here compared to the first version of the manuscript", **we have now decided to completely remove the sentence “observed isotopic variations did not necessarily follow expectations from mass-dependent fractionation”** in order to satisfy the reviewer and editor. We regret that we have to hide this information, but it is clear from the editor's decision letter that we should comply with the reviewer's view point.

Concerning the amplitude of the above-mentioned deviations from mass-dependent fractionation in these two selected samples, it is in fact much smaller than once reported in the authors own published data (Avice et al. GCA 2018, ^{124}Xe on Fig.1b, and ^{126}Xe on Fig.1c, see figure in the pdf file). The results of this study is central to the model of ‘step by step’ $\text{H}^+ - \text{Xe}^+$ escape from Earth’s atmosphere, and as such is key in linking the Great Oxidation Event to the missing Xe paradox, as proposed by the authors. If we are to discredit a paper and its review process for a limited amount of individual data points a bit off of a regression line, then the model linking the Great Oxidation Event to the missing Xe paradox has to be discredited too? Or perhaps the authors have an insight on what causes these strong deviations which could also help interpreting the smaller deviations they point out in others’ data-sets.

Fig. 1. Isotopic spectra of Xe released from fluid inclusions in samples from (a) Barberton, (b) Fortescue Group, (c) Vetryny Belt, from Avice et al. GCA 2018

Agreed but there is a wealth of discussion that already exists in the literature about the deviations observed in Avice et al. GCA 2018. The different mechanisms potentially accounting for these deviations (i.e., radioactive decay, thermal neutron capture, addition of cosmogenic Xe in the present-

day atmosphere) are discussed in Avice et al. Nat. Com. 2017 Suppl material, after insightful interactions between the first author of this study and a reviewer. Only natural samples can suffer from such effects. Therefore, these results cannot be used to explain isotope anomalies arising during the Rzeplinski experiments. Crucially, however, such details are far beyond the scope of this study, which, according to the reviewer, “has a much broader scope, with less emphasis on xenon”.

7)1376-389: ‘Most importantly, the progressive release of mantle Xe that had been repeatedly enriched in Xe relative to other noble gases (including Kr) is unable to produce an atmosphere with a Xe deficit, starting from a CI-like atmosphere.’

Element fractionation works both ways, hence if Xe is preferentially incorporated in crystals upon magma ocean crystallization or during crystals/atmosphere equilibrium under pressure, the same is true for noble gas release: Xe is expected to be preferentially retained in crystals upon further partial melting or fluid release at depth. Only in case of complete degassing would be the statement by the authors true. The authors should discuss such cases or just remove the sentence.

The authors propose that crystals formed upon magma ocean crystallization had a very low Kr/Xe ratio due to the preferential incorporation of Xe into minerals under equilibrium. Then, they suggest that it is possible to generate an atmosphere with a greater Kr/Xe than the original (chondritic) atmospheric composition via melting/degassing of minerals with low Kr/Xe, because Xe would be preferentially retained into these minerals under equilibrium. See cartoon below for illustration. In case of complete degassing, we would get the composition that was recorded in deep minerals (so a low Kr/Xe, incompatible with observations). In case of partial degassing, we would get a greater Kr/Xe in the atmosphere than what was recorded in minerals, as a result of solubility equilibrium. But, as far as we understand it after in-depth discussion between coauthors of our review paper, as well as with colleagues from our corresponding institutes, the Kr/Xe in the atmosphere would not become greater than the original (chondritic) ratio.

There might also be a confusion here about the initial chondritic atmosphere that underwent mass-dependent fractionation in the Rzeplinski et al. scenario, and the CI-like late veneer. It is the minor release of heavy trapped Xe in this CI-like late veneer atmosphere that is advocated in their scenario to explain atmospheric Archean evolution.

This is now clearly explained L371-374 “A late influx of chondritic material toward the end of Earth’s accretion would have then reset the atmosphere to a CI-like composition, which would have then evolved during the Archean through partial remobilization of heavy Xe previously trapped in deep silicate phases”. We hope that this description of their model is now appropriate.

The sentence L377-380 has also been modified for clarity: “For example, how the release of mantle Xe – repeatedly enriched in Xe relative to other noble gases (including Kr) and compared to a CI-like

starting composition – could ultimately produce an atmosphere with a Xe deficit (still relative to Kr and CI chondrites) remains to be quantitatively assessed.”

The concluding sentences “The proposal that repeated interactions between the early Earth’s atmosphere and its silicate reservoirs led to a missing and isotopically fractionated atmospheric xenon after several magma ocean episodes can therefore largely be questioned. The cause(s) of these isotopic variations remain unknown, but it leaves little doubt that they are incompatible with the mass-dependent trend shown by atmospheric xenon relative to plausible starting compositions” have simply been removed from our manuscript to satisfy the reviewer, even though they truly reflect our thinking.

8) L376-389: *‘The proposed model would also require mantle Xe to be enriched in heavy Xe isotopes relative to the atmosphere, which is opposite to observations of light Xe isotope enrichments in mantle gases worldwide.’*

No, Rzeplinski et al. do not require mantle Xe to be enriched in heavy Xe isotopes relative to the atmosphere. As explicit in the Extended Data-Fig.4, mantle Xe in their scenario has the same Xe isotopic signature as the current-day atmosphere. The authors besides forget to mention here how Rzeplinski et al. explain the observation of light Xe isotope enrichments in mantle gases, i.e. by the contribution of recycled Archean atmosphere to the mantle source or by input from less or not fractionated lower mantle resulting from the last magma ocean stage having affected mostly/only the upper mantle and lower crust; these possibilities are also illustrated in the Extended Data-Fig.4.

The sentence “*The proposed model would also require mantle Xe to be enriched in heavy Xe isotopes relative to the atmosphere, which is opposite to observations of light Xe isotope enrichments in mantle gases worldwide.*” Has now been removed from the manuscript in order to satisfy the reviewer. However, their statement is confusing. The reviewer says “mantle Xe in their scenario has the same Xe isotopic signature as the current-day atmosphere” and the model “can also explain the observation of light Xe isotope enrichments in mantle gases”. If there are light Xe isotope enrichments in mantle gas relative to the atmosphere, then the mantle is not the same as the atmosphere. So, these two statements are simply incompatible.

About archean Xe recycling: we have discussed with multiple noble gas colleagues from different laboratories worldwide, and we cannot find a way to justify such a far-fetched scenario. The reviewer is saying that the present-day mantle would be enriched in light Xe isotopes due to the subduction of Archean Xe. However, the ancient atmosphere was, according to their scenario, itself evolving due to the degassing of the mantle. How can the mantle be both (i) the cause of atmospheric evolution, and (ii) the place where ancient atmospheric compositions are recorded? This is simply not defensible and we cannot suggest otherwise in this manuscript.

This is now specified L380-384: “Rzepliński et al. (2022) suggest that the enrichment of light Xe isotopes in mantle gases may be explained by the contribution of recycled Archean atmosphere to the mantle source. However, the idea that the mantle is both (i) the driver of atmospheric evolution through degassing and (ii) the reservoir preserving ancient (Archean) atmospheric compositions appears problematic.”

9) *‘At last, why the Xe isotope evolution of Earth’s atmosphere (magmatic activity and continental erosion/metamorphism) would have stopped around the Great Oxygenation Event remains to be explained in the framework of this model.’*

This sentence implies that Xe isotope evolution abruptly stopped around the Great Oxygenation Event. However, judging from Fig.2a, it is not possible to resolve within the actual spread of data and associated error bars, if Xe isotope evolution decreased linearly throughout the Archean until to 2 Gy or if it was a parabolic decay, consistent with an equilibrium state reached around this time which also coincides with most of the continental crust being built up. Both possibilities should be discussed.

We agree with the reviewer on this point. Although we find the hypothesis of a causal link between continental crust buildup and atmospheric evolution unconvincing, we have included it (without emphasizing our disagreement) in order to address the reviewer's concern.

We now specify the following, L208-213: "Variations in atmospheric Xe isotopes through the Archean period appear to have stopped around the GOE, similar to sulfur mass-independent fractionation (S-MIF) signals (Ardoin et al., 2022). Although it is not possible to firmly establish whether Xe isotope evolution followed a linear, exponential, or power law decay throughout the Archean (Bekaert et al. 2018), these variations may provide insights into past history of continental crust build up (Rzepliński et al. 2022) and ancient levels of oxygen (O₂), methane (CH₄), and hydrogen (H₂) in the atmosphere."

10) L.389-393: The mineralogy of the upper and lower mantle are different, and so far, Xe retention has only been reported in olivine (Crepisson et al. 2018), not in bridgmanite (Shcheka and Keppler 2012). The lower mantle is therefore not expected to contain significant amounts of trapped Xe. Please be specific when mentioning deep mantle, i.e. upper mantle or lower mantle.

The "deep mantle" does not correspond to the lower or upper mantle as historically defined by mineralogists and petrologists. Rather, the "deep mantle" corresponds to pockets of deep material connected to the lowermost portions of the mantle, near the core mantle boundary, and probably associated with Large Low Shear Velocity Provinces (LLSVP). In order to avoid confusion, we now use the term "deep plume mantle sources" (L391, 393). Thank you.

11) L.637: Nitrogen can also be retained as N₂ in hydrous silicates, as shown in cymrite in metapelites for instance (Sokol et al. 2020).

The paper from this author in 2020 is "Sokol, A. G., Tomilenko, A. A., Bul'bak, T. A., Sokol, I. A., Zaikin, P. A., & Sobolev, N. V. (2020). Composition of reduced mantle fluids: evidence from modeling experiments and fluid inclusions in natural diamond. *Russian Geology and Geophysics*, 61(5-6), 663-674". We thoroughly checked and could not find a discussion about cymrite in metapelites.

One paper from these authors was published recently in *Geochemistry International*: Sokol, A. G., Korsakov, A. V., & Kruk, A. N. (2024). The Formation of K-Cymrite in Subduction Zones and Its Potential for Transport of Potassium, Water, and Nitrogen into the Mantle. *Geochemistry International*, 62(12), 1322-1331. This paper indicates that "the stability of phengite and its potential replacement by K-cymrite depends on the P-T conditions and the amount of volatiles in the metasediment." They conclude that the significance of cymrite in the overall nitrogen cycle remains to be established due to the complete dissolution of phengite with increasing P-T conditions in presence of supercritical fluid-melt, or phengite transformation into an anhydrous K-hollandite. No change to our manuscript seems required for clarity and simplicity.

Reviewer #2 (Remarks to the Author):

My apologies, I was at the Goldschmidt meeting this past week. The authors have done an excellent job revising this paper, I don't see further room for significant improvement and suggest it be published as it is. It will be highly cited and I intend to use it in my upcoming class on "origin and evolution of Earth's atmosphere."

We thank the reviewer for their positive assessment of our work and are happy that our contribution will find immediate applications.

Reviewer #3 (Remarks to the Author):

I am satisfied with the authors' responses to my overall minor comments. I have also read the other two reviewers' comments and the authors' responses. I am impressed with the breadth and depth of the conversation. However, my impression is that the authors could have avoided much of their effort in adding content on "other volatile elements" by just revising the title to focus on noble gases and N₂. After all, only noble gases and N₂ can be faithfully preserved in fluid inclusions.

We thank the reviewer for their positive assessment of our work and response to all reviewers.

Reviewer #1 (Remarks to the Author):

The revised manuscript provides a fairer analysis of the literature on the missing Xe paradox, if not an entirely unbiased one. Readers should be able to form their own opinion, provided the final points below are taken into account.

The points below have been integrated into the final version of the manuscript.

1) 1.290-292: ‘However, it is now clear that none of these hypotheses is geochemically sound (BOX 2), as they fail to explain both the depletion of Xe relative to Kr in Earth’s atmosphere, and, most critically, the isotopic evolution of atmospheric Xe throughout the Archean (Figure 2).’

This still needs to be softened up. There is no evidence here to suggest that all hypotheses concerning the storage of Xe at depth fail to explain the isotopic evolution of atmospheric Xe throughout the Archean. However, these hypotheses do face challenges, and this could be written as such.

Okay. This sentence has been reformulated as: “However, these hypotheses face great challenges in accounting for the depletion of Xe relative to Kr in Earth’s atmosphere and, most critically, the isotopic evolution of atmospheric Xe throughout the Archean (Figure 2).”

2) 1.377: ‘several’ should be changed to ‘some’, since no more than two potential caveats are discussed below (see the point below regarding the third one, which is not straightforward).

Okay, the word “several” was removed. The sentence now reads « However, this scenario entails significant caveats. » L376

3) 1.385-387: In their rebuttal letter, the authors agree that it is not possible to determine from the actual data and associated error bars whether Xe isotope evolution decreased linearly throughout the Archean until 2 Gy, or whether it decayed in a parabolic manner, which would be consistent with an equilibrium state being reached around this time, when most of the continental crust was formed. They have corrected the manuscript on lines 208–213, but not at the end of Box 2, where they repeat the argument. Either remove this argument or repeat l. 208–213.

Okay, the mention to the Great Oxygenation Event has now been removed. The text reads “Ultimately, the reason why the Xe isotopic evolution of Earth’s atmosphere eventually stopped has yet to be explained in the context of this model. » L 383-385.

4) 1.747-749: While the authors acknowledge the advantages of their escape scenario, they also recognise that it faces challenges, particularly the absence of a physical vertical transfer process. Therefore, it cannot be concluded that the missing Xe paradox has been solved. This sentence should be changed to: ‘Coupling Xe isotope analyses in ancient fluid inclusions with modelling of Xe photochemistry in the atmosphere could help to solve the missing Xe paradox, provided that a physical process for lifting Xe throughout the atmosphere is identified.’

Agreed. The text has now been revised following the reviewer's recommendation "Coupling Xe isotope analyses of ancient geological materials with models of atmospheric Xe photochemistry shall help resolve the missing Xe paradox, provided that a physical process capable of transporting Xe through the atmosphere is identified. » L745-747